# Preunderstanding, Presuppositions and Biblical Interpretation

## Thomas A. Howe

Bible and Biblical Languages, Southern Evangelical Seminary, Charlotte, NC 28277, USA; thowe@ses.edu

**Abstract:** For contemporary biblical scholars, the recognition that everyone interprets a text through one's presuppositions and preunderstanding is axiomatic. If anyone claims to approach the biblical text without any presuppositions, this is in fact a presupposition. The pervasive recognition of presuppositions and preunderstanding in interpretation has largely developed out of the influence of modern philosophy, particularly in such representatives as Immanuel Kant, Martin Heidegger, and Hans-Georg Gadamer. One's presuppositions and preunderstanding form the grid through which one interprets everything, not only texts. The pervasiveness of presuppositions and preunderstanding has issued in a wholesale rejection of the possibility of objectivity in interpretation. This essay will argue that the rejection of the possibility of objectivity is self-defeating.

**Keywords:** presuppositions; preunderstanding; hermeneutics; interpretation; Gadamer; Heidegger; objectivity

## 1. Introduction

It seems axiomatic that theology must derive from Scripture. Of course this necessarily involves the interpretation of the text. It is also held to be axiomatic that no one interprets the text apart from his preunderstanding and presuppositions. The question of the role of preunderstanding and presuppositions in understanding has become a dominant topic in contemporary biblical hermeneutics. This emphasis is particularly the impact on contemporary biblical hermeneutic theorists of the philosophies of Immanuel Kant (1724–1804), Martin Heidegger (1889–1976), and Hans-Georg Gadamer (1900–2002). It is a universally held position that interpretation necessarily involves the presuppositions and preunderstanding of the interpreter. If anyone claims to understand anything apart from presuppositions, this in itself functions as a presupposition. Typical of this position is the following assertion by the authors of a popular hermeneutics textbook on biblical interpretation; "No one interprets in a vacuum; everyone has presuppositions and preunderstandings . . . No one comes to the task of understanding as an objective observer. All interpreters bring their own presuppositions and agendas, and these affect the ways they understand as well as the conclusions they draw." (Klein et al. 2017, pp. 44–45). The problem arises from the fact that hermeneutic theorists argue that one's preunderstanding and presuppositions obviate the possibility of objectivity.in interpretation. However, such claims are ultimately self-defeating.

## 2. The Pervasiveness of Presuppositions

An acknowledgment of the all-pervasiveness of presuppositions in interpretation has virtually become a precondition for any rational discussion of the nature of understanding and interpretation. Richard Palmer asserts, "There can be no 'presuppositionless' interpretation. A biblical, literary, or scientific text is not interpreted without preconceptions. Understanding, since it is an historically accumulated and historically operative basic structure, underlies even scientific interpretation . . . Inside or outside the sciences there can be no presuppositionless understanding." (Palmer 1969, p. 182). Palmer's assertion takes its cue, as he acknowledges, from the title of Rudolf Bultmann's famous article, "Is Presuppositionless Exegesis Possible?"[1] and follows the exposition of Gadamer. Both

Gadamer and Bultmann identify the seminal work of Heidegger who proposed the view that all interpretation necessarily involves preconditions, as he asserted in *Sein und Zeit*: "Furthermore, all interpretation moves in the marked pre-structure. All interpretation that is intended to provide understanding must already have understood what is to be interpreted. This fact has always been noted, if only in the realm of derived modes of understanding and interpretation, in philological interpretation."[2]

The inevitability of presuppositions in interpretation has likewise become a recurring theme in contemporary Evangelical hermeneutics. As we have seen, William Klein, Craig Blomberg, and Robert Hubbard assert that no one interprets anything without a set of underlying assumptions:

> When we presume to explain the meaning of the Bible, we do so with a set of preconceived ideas or presuppositions. These presuppositions may be examined and stated, or simply embraced unconsciously. But anyone who says that he or she has discarded all presuppositions and will only study the text objectively and inductively is either deceived or naive. So as interpreters we need to discover, state, and consciously adopt those assumptions we can agree to and defend, or we will uncritically retain those we already have, whether or not they are adequate and defensible (Klein et al. 2017, p. 87).

According to Klein, Blomberg, and Hubbard, no only is the interpretation of the Bible subject to the presuppositional perspective of the interpreter, but all inquiry necessarily requires presunderstanding. As they go on to say, "we argue simply that an appropriate level of preunderstanding is necessary for any kind of knowledge. This, as we have seen, is the nature of all inquity" (Klein et al. 2017, p. 114).

In his book on Old Testament interpretation, John Goldingay acknowledges that the study of the Bible necessarily involves the presuppositions of the interpreter: "It is actually impossible to study without have [sic] one's own beliefs and framework of thinking, and being influenced by them. Indeed, we need some such framework if we are to make coherent sense of the data we examine. What is important is to be open to recognizing our presuppositions, and then to be prepared for the material we are studying to challenge them and to modify the perspective with which we approached it." (Goldingay 1990, p. 18).

In his examination of the inductive method of Bible study, Elliott Johnson asserts that the inductive method of Bible study is inadequate because it does not take account of the interpreter's presuppositions that are necessarily involved in the study of the biblical text. Johnson believes inductive methods fail "because there is no such thing as 'pure' inductive study. We all necessarily bring premises or presuppositions to the study of the text. Those premises affect the way we comprehend the meaning, the way we understand—that is, they have epistemological influence. This influence in the interpreter has been at the center of hermeneutical concerns since the time of Immanuel Kant (1724–1804)." (Johnson 1990, p. 19).

In the volume that heads the series *Foundations of Contemporary Interpretation*, Moisés Silva made the same kind of observation. After a brief overview of the influence of recent philosophy on hermeneutical thinking, Silva observes that the influence of the phenomenology of Martin Heidegger has forced evangelical theorists to acknowledge the role of preunderstanding in all kinds of thinking. In Silva's words: "This idealist tradition is vulnerable to some powerful criticisms, yet within the context of that tradition some of the most crucial questions about hermeneutics have arisen. Such thinkers as Martin Heidegger, for example, have forced us to take seriously the role that *preunderstanding* plays in the process of interpretation. None of us is able to approach new data with a blank mind, and so our attempts to understand new information consist largely of adjusting our prior 'framework of understanding'—integrating the new into the old." (Silva 1987, p. 6).

Silva goes on to question the possibility of presuppositionless interpretation when he says, "The common insistence that we should approach the text without any prior ideas regarding its meaning becomes almost irrelevant . . . Could it be that it is impossible to

shed our presuppositions precisely because it is they that mediate understanding?" (Silva 1987, pp. 6–7).

Although Grant Osborne uses the terms 'preunderstanding' and 'presupposition' almost synonymously, he asserts the same notion that the inevitability of one's preunderstanding provides the perspective for understanding: "A close reading of the text cannot be done without a perspective provided by one's preunderstanding as identified by a 'sociology of knowledge' perspective. Reflection itself demands mental categories, and these are built upon one's presupposed world view and by the faith or reading community to which one belongs. Since neutral exegesis is impossible, no necessarily 'true' or final interpretation is possible." (Osborne 2006, p. 516).

Finally, Dan McCartney and Charles Clayton make the same point. They assert that the interpreter must know himself and his own presuppositions and how they influence his understanding of the Bible: "Interpreting any text involves two different types of assumptions. First, underlying all our thinking and interpreting are our presuppositions about life and ultimate realities, our worldview. These provide the basic foundation for how we understand everything. Second are the assumptions which we make about the nature of the text we are reading." (McCartney and Clayton 1994, p. 13).

The realization of the fundamental role of presuppositions in interpretation is not entirely new. In his work, *Hermeneutics of the New Testament*, published in English in 1877, Albert Immer addressed the question of the perspective of the interpreter: "Lastly, (d) the interpreter has to remove the difference between the *work* of the author and his own view of the matter. The work of the author in its totality is just a historical fact, and is to be treated as such. In this, the more perfectly the interpreter can abstract himself from his own opinions or knowledge, and, by virtue of his historical information, can throw himself into his author and his time, the more successful he will be."[3] In his massive work on interpretation published in 1883, Milton Terry recognized the place of "presumptions" in biblical hermeneutics. He said, "The question should be, not what does the Church say, or what do the ancient fathers and the great councils and the œcumenical creeds say, but what do the Scriptures legitimately teach? Still less should we allow ourselves to be influenced by any presumptions of what the Scriptures *ought* to teach." (Terry 1883, p. 595). It seems that both Immer and Terry are advocating the very approach that contemporary hermeneutic theorists declare is not possible.

The assertion by Milton Terry sounds quite similar to the assertion that would later be made by Rudolf Bultmann. In his article on presuppositions and exegesis, Bultmann asserted that exegesis must be presuppositionless in the sense that "it not presuppose (*voraussetzt*) its results (*Ergebnisse*)." (Bultmann 1957, p. 409). However, Bultmann goes on to assert that the condition of being absolutely presuppositionless is not possible. He declared, "The question of exegesis without presuppositions in the sense of unprejudiced exegesis must be distinguished from this same question in the other sense in which it can be raised. And in this second sense, we must say that *there cannot be any such thing as presuppositionless exegesis*. That there is no such exegesis in fact, because every exegete is determined by his own individuality, in the sense of his special biases and habits, his gifts and his weaknesses . . . "[4]

Anthony Thiselton prefers the term 'horizon of expectation' to the traditional term, 'preunderstanding'. Nevertheless, Thiselton affirms the inevitability of the horizon of expectation as an unavoidable precondition of interpretation. He says, "First, [horizon of expectation] includes a network of provisional working *assumptions* which are open to revision and change; second, the reader or interpreter may not be *conscious* of all that the horizon of expectation sets in motion, makes possible, or excludes. The expectations and assumptions concern the kind of questions and issues which we anticipate the text will address, and even the types of genre or mode of communication which it might use." (Thiselton 1992, p. 44).

Thiselton's description of the horizon of expectation as distinguished from "preunderstanding" is further distinguished from "presupposition." Whereas the term 'presup-

position' seems to suggest "the impression of rooted beliefs and doctrines which are not only cognitive and conceptual, but which also can only be changed and revised with pain, or at least difficulty," (Thiselton 1992, p. 45). the term 'horizon of expectation,' or simply 'horizon,' avoids the conceptual baggage and implies one's "relation to *life* . . . My pre-conceptual relation in life to music, to mathematics, to law, or to love" which "both influence, and provide operative conditions for, how I came to understand texts of music, mathematics, law, poems or letters or declarations of love." (Thiselton 1992, p. 45). Ultimately, Thiselton's description of the term 'horizon' indicates that he conceives horizon in a very broad sense, roughly equivalent to what has become popularly known as a world view.

> The horizon or pre-intentional background is thus a network of revisable expectations and assumptions which a reader brings to the text, *together with* the shared patterns of behaviour and belief with reference to which processes of interpretation and understanding become operative. The term "horizon" calls attention to the fact that our finite situatedness in time, history, and culture defines the present (though always expanding) limits of our "world", or more strictly the limits of *what we can "see"*. The term "background" calls attention to the fact that these boundaries embrace not only what we can draw on in conscious reflection, but also the pre-cognitive dispositions or competencies which are made possible by our participation in the shared practices of a social and historical world (Thiselton 1992, p. 46).

Whether we employ the terms, 'presupposition' or 'preunderstanding,' or whether we adopt the broader terminology of Thiselton, it is generally agreed by contemporary hermeneuticians that some form of pre-condition not only precedes the act of interpretation, but that this pre-condition is the means by which interpretation is possible. As Roger Lundin asserts, "Rather than standing as impediments to right understanding, then, 'prejudices' are the basis of all understanding. The goal of all thinking—and reading—should not be to cast these assumptions aside in order to take on those belonging to another person . . . Rather, the goal of thinking should be to test, clarify, modify, and expand our assumptions, to bring them more in line with a more comprehensive truth." (Lundin 1997, p. 162).

Lundin provides a helpful summary of the prevailing perspective of contemporary hermeneutical theory with respect to the role of presuppositions, prejudices, or horizon: " . . . modern hermeneutical reflection calls into question the Cartesian and Enlightenment claims for the supremacy of skeptical detachment and rational reflection. On a practical level, hermeneutics has questions whether such complete detachment is a possibility. Do we ever read with all our assumptions held in abeyance? Do we completely set aside our beliefs when we enter the alien world of a text? To such questions, hermeneutical theory responds with a convincing 'No'." (Lundin 1997, p. 162).

This small sampling of contemporary evangelical texts on hermeneutics should not mislead. These examples represent an almost universally held assumption about the task of hermeneutics, namely, that interpretation necessarily and inevitably operates from the perspective of the presuppositions of the interpreter.[5] In other words, it would seem that the presuppositions of the interpreter are the omnipresent grid through which he or she interprets or understands anything.

The idea that knowledge and interpretation inevitably involve the presuppositions of the interpreter seems to be an inescapable conclusion. It seems not only to be the case that the assumption of presuppositionless interpretation is a naive approach, but even the idea that one can approach the text without presuppositions is itself a presupposition.

### 3. The Mutability of Presuppositions

There is another notion that is no less universally affirmed by contemporary hermeneutic theorists. Although presuppositionless interpretation is understood to be impossible, according to the contemporary wisdom, this does not mean that interpretation must be static or without growth. Rather, growth in understanding is possible because an inter-

preter should be able to change his presuppositions. Contemporary theorists maintain the possibility and indeed the necessity of altering, modifying, and developing the preconditions, presuppositions, prejudices, or horizons of expectation that make understanding possible. As Palmer asserts, "The point is that our own presuppositions must not be taken as absolute (for these are the foundation of our expectations) but as something subject to change . . . What is needed in literary interpretation is a dialectical questioning which does not simply interrogate the text but allows the thing said in the text to interrogate back, to call the interpreter's own horizon into question and to work a fundamental transformation of one's understanding of the subject." (Palmer 1969, pp. 233–34).

Even though our presuppositions, or prejudices, or horizons of understanding are inevitable, according to the prevailing assumption, they are not immutable. The interaction of the interpreter with the text creates the possibility that the text can impact the interpreter so as to cause the interpreter to modify his pre-conditions of interpretation. Indeed, the commitment of contemporary Evangelical hermeneutic theorists to the mutability of presuppositions and preunderstanding is an immutable assumption.

By acknowledging the inevitability of the pre-conditions of interpretation, Evangelicals have not thereby abandoned the hope of adjudicating between interpretations, as is evidenced in the following claims of McCartney and Clayton.

We will argue later that there is a right way to understand Isaiah 53 or any other passage and that the right way is indicated by the nature of the text itself. However, discerning this is not a matter of escaping or suspending our presuppositions, but changing and adapting them. We really cannot escape them. Since the things we assume are to us self-evident, we may be unconscious of them, but they still determine our understanding, and without them there is no understanding. Any time we find "meaning" in a text, we arrive at that "meaning" by fitting it in with our previous knowledge. And this involves assumptions or presuppositions about such things as the nature of the text we are reading, the meaning of life, and how we know things. All our interpreting activity in life involves assumptions, just as in geometry every theorem can only be proven on the basis of previous theorems and "self-evident" assumptions. Presuppositions form the basis of the "interpretive framework" by which we understand things. (McCartney and Clayton 1994, p. 14).

Although McCartney and Clayton hold that "all our interpreting activity in life involves assumptions," they still claim that it is possible to change and adapt these preconditions as the interpreter encounters the text. Indeed, it is not only possible to change and adapt one's presuppositions, but it is possible to eliminate some presuppositions altogether. Other Evangelicals acknowledge the mutability of preunderstanding as well: "The honest, active interpreter remains open to change, even to a significant transformation of preunderstandings." (McCartney and Clayton 1994, p. 115).

## 4. Basic, Fundamental, and Foundational Presuppositions in Hermeneutic Thinking

Along with the inevitability and mutability of presuppositions, there is a third ingredient that is important in understanding the contemporary view of the relation of presuppositions and preunderstanding to interpretation. Several hermeneutic theorists have made reference to "basic" or "fundamental" assumptions or presuppositions. Palmer refers to understanding as "an historically accumulated and historically operative basic structure" that "underlies even scientific interpretation." (Palmer 1969, pp. 182–83). Ted Peters refers to presuppositions deriving from "a more basic or foundational level of one's way of viewing things." (Peters 1974, p. 210). He refers to what he calls "fundamental presuppositions" which "refer us to the basic structure of one's experience with the world" which "makes articulation of that experience possible." (Peters 1974, p. 210). Peters even refers to the existence of fundamental presuppositions, self-evident truths, and axioms. (Peters 1974, p. 211).

Similar descriptions are found among Evangelical theorists. McCartney and Clayton identify "presuppositions about life and ultimate realities, our worldview" as providing "the basic foundation for how we understand everything." (McCartney and Clayton 1994,

p. 13). Robertson McQuilkin discusses the "basic presupposition about the Bible that distinguishes believers from unbelievers." (McQuilkin 1992, p. 19). Vern Poythress talks about "basic commitments or presuppositions in the formation of knowledge" (Poythress 1988, pp. 121–22). which are "fundamental assumptions about the nature of the world." (Poythress 1988, p. 122). Poythress identifies some presuppositions as "unconsciously assumed elements of one's world view" which constitute "one's basic commitments," (Poythress 1988, p. 164). and which are part of the interpreter's presuppositions or world view, and include "fundamental assumptions about the world and reality," and that these are found at the "fundamental level." (Poythress 1988, p. 122). V. Phillips Long also refers to "basic intellectual and spiritual commitments ('how he or she sees the world')." (Long 1994, p. 120).

Theorists repeatedly employ the terms 'basic,' 'fundamental,' and 'foundational' in attempting to identify that "something" beyond which there is no need to go and upon which understanding and interpretation is built. Thiselton also refers to an interpreter's horizons as "background" which makes understanding possible. (Thiselton 1992, p. 46). Horizons are pre-intentional and pre-cognitive dispositions and competencies that provide the basis upon which one participates in the social and historical world.

Many theorists even go so far as to use the term 'foundational.' Peters asserts that there is some foundational level from which presuppositions derive. (Peters 1974, p. 210). Palmer declares that although presuppositions "must not be taken as absolute" they nevertheless "are the foundation of our expectations." (Palmer 1969, pp. 233–34). With many seeking to jettison foundationalism, theorists cannot seem to do without a foundation.

One crucial omission by virtually all theorists becomes obvious from this review, however. This omission is a complete lack of any claims regarding the origin of these basic, fundamental, and foundational presuppositions. If all interpretation takes place "only in light of previous commitments," (Kaiser and Silva 1994, p. 242) as Silva (along with virtually all hermeneuticians) asserts, then from where do these "previous commitments" come?

Concerning prejudices Gadamer writes,

> . . . it is not so much our judgments as our prejudices that make up our being . . . Prejudices are not necessarily unjustified and erroneous so that they obscure the truth. In truth, it is inherent in the historicity of our existence that prejudices, in the literal sense of the word, constitute the prior directionality of all our ability to experience. They are presuppositions of our openness to the world, which are downright conditions for us to experience something, for what we encounter to say something to us (Gadamer 1993)[6].

Gadamer goes on to explain, "Only such an acknowledgment of the inherent nature of the prejudice of all understanding sharpens the hermeneutic problem to its real point."[7]

Basically, Gadamer's prejudice has taken over the place of Descartes' innate ideas or Kant's categories and constitutes the grid through which a text is made intelligible. Richard Bernstein concludes that, "If Gadamer is right in claiming that not only understanding but all knowing 'inevitably involves some prejudices,' then it is difficult to imagine a more radical critique of Cartesianism, as well as of the Enlightenment conception of human knowledge." (Bernstein 1983, pp. 127–28).

On the contrary, however, if Gadamer is in fact right, then it is difficult to imagine a view more consistent with the very Cartesian/Kantian conception of human knowledge than that which Gadamer proposes. Gadamer has simply translated the Cartesian innate ideas and the Kantian categories into Heideggerian prejudices which constitute our being. It seems to be the same idealism in existential/phenomenological clothes. Instead of referring to innate ideas or categories of the mind, one simply talks about metaphors and presents the appearance of having solved the problem of knowledge. Nevertheless, the problem of the foundations of knowledge still remains.

Bernstein argues that, "in these [Cartesian and Enlightenment] traditions there are sharp dichotomies between reason and prejudice, or between knowledge and prejudice." (Bernstein 1983, p. 128). Another basic problem with Cartesian and Enlightenment episte-

mology is the conception of the nature of reason itself and the starting point that constitutes the Cartesian foundation of knowing. Bernstein argues, following Charles Peirce and Gadamer, that, "There is no knowledge without *pre*conceptions and *pre*judices. The task is not to remove all such preconceptions, but to test them critically in the course of inquiry." (Bernstein 1983, pp. 127–28).

### 5. Presuppositions and the Hermeneutical Circle

There begins to emerge a relationship between the unavoidable preconditions of interpretation and the foundational nature of at least some of our presuppositions on the one hand, and the circularity of the hermeneutic enterprise on the other. Although, according to the current wisdom, there is no interpretation apart from the presuppositions and prejudice of the interpreter, these presuppositions and prejudices that form the preconditions of interpretation are not beyond scrutiny. According to the predominate understanding, presuppositions and prejudices can be measured against the world in a circular or spiral, dialectical motion.

Peters, for example, applies the principle of criticism to understanding in terms of the hermeneutical circle: "What is required for an understanding of texts, or of one's tradition, is some sort of pre-apprehension or pre-understanding of the whole of which the objects of study are parts . . . In order to interpret anything, we must begin by projecting a preunderstanding of what it is we are about to interpret." (Peters 1974, p. 212). Peters has applied the hermeneutical circle to understanding itself. If presuppositionless exegesis is not possible, at least in the sense that everyone comes to the text with convictions, points of view, experiences, assumptions, and a philosophical framework, then a point of urgent consideration is how one can be sure that one's presuppositions and preunderstanding facilitate interpretation rather than hinder it.

The inevitability and mutability of presuppositions has given new significance to the old problem of the hermeneutical circle. Friedrich Schleiermacher attributes the articulation of the hermeneutical circle to his predecessor Friedrich Ast (1778–1841) when he says, "The hermeneutical principle put forward by Mr. Ast and, in some respects, fairly elaborated, that although the whole can be understood from the detail, the detail can only be understood from the whole, is of such scope for this art and so undeniable that that even the first operations cannot be accomplished without using it, yes, that a large number of hermeneutic rules rest more or less on it."[8] The hermeneutical circle, according to Schleiermacher, can best be illustrated by the relationship of words to a sentence. A complete sentence provides the contextual framework within which the parameters are set with respect to the possible meanings of the individual words. However, it is the contribution of each individual word that constitutes the meaning of the complete sentence. "The circle as a whole defines the individual part, and the parts together form the circle." (Palmer 1969, p. 87). So, the individual concepts that are expressed in any literary piece, whether they be written or oral, are determined by the context in which they stand. Yet, the context is constituted by the combination of the individual concepts. "By dialectical interaction between the whole and the part, each gives the other meaning; understanding is circular, then. Because within this 'circle' the meaning comes to stand, we call this the 'hermeneutical circle.'" (Palmer 1969, p. 87).

With the contemporary understanding of the all-pervasiveness and mutability of presuppositions, the hermeneutical circle, which at one time was discussed in terms of the text, now applies to all of life. As Brice Wachterhauser describes it, "The hermeneutical circle involves the 'contextualist' claim that the 'parts' of some larger reality can be understood only in terms of the 'whole' of that reality, and the 'whole' of that reality can be understood only in terms of its parts. This is to say that understanding any phenomenon means, first of all, situating it in a larger context in which it has its function and, in turn, it also means letting our grasp of this particular phenomenon influence our grasp of the whole context." (Wachterhauser 1986, p. 23).

Wilhelm Dilthey applied the concept of the hermeneutical circle to the question of historical understanding. Not only is the record of the event or historical situation that one is endeavoring to understand located in its own historical setting, but the interpreter is likewise operating from his own historical setting. As Peters describes, "Interpretation, then, is always an event taking place in a situation, the context in which the interpreter and the text or any other expression of life stand. Meaning is, therefore, always meaning in relationship." (Peters 1974, p. 212). The impact upon hermeneutics of the broadening out of the hermeneutic circle to encompass lived experience is that understanding itself is circular. As Palmer observes, "there is really no true starting point for understanding, since every part presupposes the others. This means that there can be no 'presuppositionless' understanding.' Every act of understanding is in a given context or horizon; even in the sciences one explains only 'in terms of' a frame of reference." (Palmer 1969, pp. 120–21). Consequently, the circle of life experience disallows the possibility of a transcendental, or a-historical perspective—a view from nowhere. "Since we understand always from within our own horizon, which is part of the hermeneutical circle, there can be no nonpositional understanding of anything." (Palmer 1969, p. 121).

With Martin Heidegger the hermeneutical circle takes on proportions that extend it to the very nature of reality. He argued that all interpretation must already understand, in some sense, that which is being interpreted: "In every understanding of the world, existence is understood with it, and vice versa. All interpretation, moreover, operates in the fore-structure (*Vor-struktur*), which we have already characterized. Any interpretation which is to contribute to understanding, must already have understood what is to be interpreted."[9]

Heidegger expands the traditional view of the hermeneutic circle to encompass the very being of *Dasein*. Traditionally, the hermeneutic circle assumed a distinction between subject and object in which the knowing subject stood over against the known object. As David Hoy describes, "Heidegger conceives of Dasein and world as forming a circle, and he thus extends the traditional hermeneutic circle between a text and its reading down to the most primordial level of human existence." (Hoy 1993, p. 172).

Note, however, that the circle, for Heidegger, is not a vicious circle. He contends, "But to see a viciousness in this circle and to look for ways to avoid it, even to 'feel' [emfinden] it as an inevitable imperfection is fundamentally to misunderstand understanding."[10] Heidegger asserts, "The 'circle' in understanding belongs to the structure of meaning, which phenomenon is rooted in the existential constitution of existence, in interpretive understanding."[11] The goal is not to attempt to escape the circle, "but to enter into it in the right way."[12] This means that the interpreter should not allow his fore-structure to be dictated by arbitrary habits or fancies, but he must concentrate on "the things themselves."[13] In this way, the circle involves a dialectic or conversation, as it were, constantly to revise one's fore-structure of understanding in terms of the things themselves as the interpreter moves toward a better understanding.

Gadamer adopted Heidegger's conception of the hermeneutical circle and, along with Dilthey's emphasis on historical situatedness, once again applied it to the interpretation of texts. As Gail Soffer explains,

> Gadamer explicitly applies the Heideggerian hermeneutical circle—intended as a completely general analysis of human existing—to the specific activity of interpreting texts. According to this application, when one reads a text the fore-structures (also named 'prejudices' or prejudgments [*Vorurteile*] by Gadamer, in his polemic against the Enlightenment) results in an anticipatory projection of the meaning of the whole text. However, as in Heidegger, there is a dialectical relationship between fore-structures and projected meaning." (Soffer 1992, p. 234).

Gadamer understands this dialectical process in terms of a conversation with the text. As he explains,

> That every revision of the preliminary design has the possibility of throwing out a new design of meaning, that rival designs can be brought together to be worked out side by side until the unity of meaning is more clearly established; that the interpretation begins with preliminary concepts that are replaced by more appropriate concepts: precisely this constant redesigning, which constitutes the meaning movement of understanding and interpretation, is the process that Heidegger describes. Anyone who seeks to understand is exposed to the confusion of preconceptions that do not prove themselves in the things themselves. The constant task of understanding is the elaboration of the right, appropriate drafts, which as drafts are anticipations that are only to be confirmed "in the things."[14]

So, it would seem that the existence of the hermeneutical circle is predicated upon a broad conception of the preconditions of understanding. These preconditions project a pre-understanding of what is to be interpreted. As the interpreter interacts with the text, as it is in itself, certain prejudices are altered and even replaced by those that are "more suitable" to the emerging understanding of the text. As Ted Peters explains, "Whenever we attempt to interpret an event in our life, it is already in some sense understood; and from there we attempt to explicate or articulate it as it is found in that already understood character." (Peters 1974, pp. 213–14).

Indeed, as we have seen, all interpretation proceeds upon the basis of the pre-conditions of interpretation, and, as Moisés Silva indicated, it is our presuppositions that mediate understanding. This is precisely the perspective that has been adopted in Evangelical theories. There is perhaps no better example of this view than the position that has been argued by KBH: "Every interpreter begins with a preunderstanding. After an initial study of a Biblical text, that text performs a work on the interpreter. His or her preunderstanding is no longer what it was. Then, as the newly interpreted interpreter proceeds to question the text further, out of this newly formed understanding—perhaps, different—answers are obtained. A new understanding has emerged. It is not simply a repetitive circle; but, rather, a progressive spiral of development." (Klein et al. 2017, pp. 113–14).

By way of illustration, KBH offer the diagram in Figure 1 (see page 27). The starting point for understanding is within the field of the preunderstanding of the interpreter. As the interpreter interacts with the text, the text is allowed to have its influence on the preunderstanding of the interpreter, and the action moves back and forth from interpreter to text and from text to interpreter. KBH acknowledge that their analysis admits "an inevitable circularity in interpretation." (Klein et al. 2017, p. 114). They seem to sense a viciousness to this circularity and attempt to blunt this implication:

> When we posit the requirement of faith to understand the Bible fully and then we go to the Bible in order to understand God's self-revelation in Christ in whom we have faith, the process has a definite circularity. But we argue simply that an appropriate level of preunderstanding is necessary for any kind of knowledge. This, as we have seen, is the nature of all inquiry. Thus, one must have some knowledge of God even to arrive at the preunderstanding of faith. Then that stance of faith enables the Christian to study the Bible to come to a deeper understanding of God and what the Scriptures say. As we learn more from our study of Scripture we alter and enlarge our preunderstanding in more or less fundamental ways. In essence, this process describes the nature of all learning: it is interactive, ongoing, and continuous (Klein et al. 2017, pp. 113–14).

In the view of KBH the circle, or spiral, is predicated on the "appropriate level" of preunderstanding. Unfortunately, there is no attempt to define what is considered an "appropriate level." This is not an isolated perspective peculiar to these particular theorists, however. This seems to be a view of the role of presuppositions and preunderstanding in hermeneutical methodology that is almost universally held.

William Larkin asserts that the hermeneutical circle is actually "a hermeneutical spiral in which interpreters through successive exposure to God's Word are able to bring their

preunderstanding and, as a result, their interpretation and application closer and closer to alignment with Scripture's truth." (Larkin 1993, p. 302).

Grant Osborne offers the following explanation: "The text itself sets the agenda and continually reforms the questions that the observer asks of it." (Osborne 2006, pp. 417–18). At this point, Osborne does not discuss the starting point by which one enters the circle, or spiral. Earlier he discusses the relationship of meaning and understanding, and one can assume that it is by virtue of one's preunderstanding that questions are formed which are addressed to the text. The text, then, acts on the interpreter who is able to refine his questions, and the spiral continues.

McCartney and Clayton offer the same estimation of the nature of the relation between presuppositions and interpretation, albeit without the benefit of a diagram:

> We can think of this as a "hermeneutical spiral" or a "spiral of understanding." Although one must know the forest in order to understand the trees, it is also true that a knowledge of the trees builds up the understanding of the forest. Our presuppositions about the overall meaning of the Bible, and life in general for that matter, form the interpretive framework for understanding particular texts of the Bible, which in turn act as a corrective to the overall interpretive presuppositions. This continual interaction moves us up a spiral toward a "meeting of meaning" and understanding of the truth (McCartney and Clayton 1994, pp. 17–18).

Among Evangelical and non-evangelical theorists, it is generally held as virtually axiomatic that presuppositions and preunderstanding constitute the preconditions for interpretation. It is also generally held that if the interpreter comes to the object of his investigation with a willingness to interact with the text-in-itself, then this dialectical or conversational interaction can serve to shape the interpreter's preconditions into a greater conformity with the object of interpretation.

It is perhaps instructive that, amid the constant assertion that one's presuppositions form the foundation and possibility of interpretation, that this very framework can be changed by interaction with the text, and yet no theorist proposes that the assumption that all interpreters approach a text with presuppositions is itself subject to change. This assumption is never the object of change. An interpreter can apparently change any other presupposition he might hold, but he cannot change the presupposition that he always approaches the text with presuppositions. Additionally, the preunderstanding of the nature of understanding is not the object of change either. Yet cannot the question be asked, What if the text proposes a scenario of understanding that is contrary to the notion that understanding is a spiral? What if the text were to assert that understanding is linear, or entirely circular? Theorists do not entertain the possibility that the nature of understanding can be anything but a spiral. This assumption is never questioned, nor is the possibility entertained that a text might seek to change this scenario. It would seem that one must ask, Why are not these presuppositions about the nature of understanding subject to change? Are not these presuppositions just as much a part of the preunderstanding of these interpreters as any other presupposition? Perhaps not all presuppositions are mutable after all.

## 6. The Influence of Hans-Georg Gadamer

Gadamer has had a huge impact on contemporary biblical hermeneutics. In her study on the exegesis of Job from Calvin to the modern era, Susan Schreiner claims that Gadamer's arguments against objectivity in interpretation have been very persuasive. If the historian cannot be free of "prejudices," then the historian cannot be assured that the author's intent can be known. Presuppositions thus become preconditions for interpretation. As Schreiner describes:

> No one has done more to shake the confidence of historians in the old ideal of objectivity than Gadamer. As Gadamer explained, the goal of Romantic hermeneutics to find authorial intention presupposed the Enlightenment ideal of a mind free from prejudices. Historians were to enter the mind of the author and to

transpose themselves into the culture of an earlier age. To this presupposed ideal of objectivity and historical empathy, Gadamer opposed the historicity of understanding. Rejecting the "prejudice against prejudice" inherited from the Enlightenment, Gadamer argued that readers cannot free themselves from their prejudices and thereby recover the mind of the author. Such shedding of presuppositions is neither possible nor desirable. Presuppositions or prejudgments are the necessary preconditions for understanding (Schreiner 1994, pp. 10–11).

The idea is that the objective historian assumes that he can ignore the inescapablility of one's perspective and its relation to the problem of objectivity in historical knowledge. George Iggers has pointed out, "Historicism has too many meanings to be useful as a term without careful delimitation. (Iggers 1983, p. 4). Jean Grondin provides a characterization of historicism in the context of Gadamer's philosophical hermeneutics: "Historicism . . . is the central and most crippling problem facing philosophy since Hegel, namely, the question concerning the possibility of binding truth and thus conclusive philosophy within the horizon of historical knowledge. Are all truths and rules of conduct dependent on their historical context? If so, the specter of relativism and nihilism lurks nearby." (Grondin 2001, p. 24). According to this view, historical phenomena can be understood only in terms of one's own historical situatedness. Each person is located in a specific place, at a specific time, in a specific culture, with certain social, political, philosophical, theological, etc., perspectives through which he or she perceives and interacts with the world around and within.

Consequently, the problem of historicism arises from the fact that all truths are dependent on the historical context of the one asserting the truth. Historical phenomena can be understood only in terms of one's own historical situatedness, or in terms of one's own place within the flow of historical development. But if any phenomenon can be understood only in terms of its own historical situatedness, this would imply that the historical situatedness of the interpreter binds him to his own place within the flow of historical development. Therefore the interpreter cannot lay claim to an a-historical, transcendent perspective. If the interpreter cannot lay claim to an a-historical, transcendent perspective, then the claim that no historical phenomenon can be understood apart from its own historical situatedness cannot itself be an a-historical, transcendent claim about historical knowledge. The historicist assertion turns out to be just another instance of self-defeating relativism. The self-referential nature of the claim of historicism indicates its self-destructive nature.

Gadamer strongly objects to arguments such as these which appeal to a self-referential problem in historicism:

> However clearly one demonstrates the inner contradictions of all relativist views, it is as Heidegger has said: all these victorious arguments have something of the attempt to bowl one over. However cogent they may seem, they still miss the main point. In making use of them one is proved right, and yet they do not express any superior insight of value. That the thesis of skepticism or relativism refutes itself to the extent that it claims to be true is an irrefutable argument. But what does it achieve? The reflective argument that proves successful here rebounds against the arguer, for it renders the truth value of reflection suspect. It is not the reality of skepticism or of truth-dissolving relativism but the truth claim of all formal argument that is affected.[15]

What Gadamer is saying is that, notwithstanding the inevitable relativism, historicism is absolutely and indubitably inescapable, and that this is a-historically and transcendentally true for all people, at all times, in all cultures. Gadamer appears to have access to an a-historical, transcendent perspective on historicism that he disallows for everyone else. What follows from this brand of historicism is ably articulated by Georgia Warnke: "The claim is that we are always involved in interpretations and that we can have no access to anything like 'the truth' about justice, the self, reality or the 'moral law.' Our notions of these 'truths' are rather conditioned by the cultures to which we belong and the historical

circumstances in which we find ourselves. Hence, we must face the fact of our finitude and the utterly contingent character of our efforts to understand." (Warnke 1987, p. 1).

Nevertheless, since the claims of historicism are indeed self-refuting, and since, as Gadamer acknowledges, "the thesis of skepticism or relativism refutes itself to the extent that it claims to be true is an irrefutable argument," then the implications of Gadamer's historicism do not follow, and absolute truth, including absolute concepts of justice, the self, reality, and moral law, does exist and is accessible by the finite mind. In fact, the existence of absolute truth is asserted by all those who advocate the absolute truth of historicism.

Gail Soffer has demonstrated the same self-referential problem with Gadamer's historicism: " . . . [Gadamer's] philosophy remains mired in the epistemological paradox, at once asserting and denying the possibility of knowledge of the unconditional. If the historicity thesis is true, then it is impossible for us to have knowledge of unconditional truths, and so of the historicity thesis."[16] Grondin proposes that both Gadamer and Heidegger have defused the self-referential argument of absolute truth:

> The philosophical achievement of hermeneutics is perhaps not so much a solution to its problem as a farewell to historicism. With Heidegger and Gadamer, historicism is applied to itself, so to speak, and thus made visible in its own historicity, namely in its secret dependence on metaphysics. Because the dogmatic thesis that everything is relative can only make sense against the horizon of an unrelative, absolute, timeless, metaphysical truth. Only on the scale of an absolute truth believed to be possible can an opinion be considered merely relative.[17]

Besides the fact that this assertion assumes that metaphysics must be illegitimate, Grondin's thesis does not work since it is not necessary to "suppose" absolute truth. Rather, absolute truth is unavoidable, not merely presupposed. Grondin asks, "What does this absolute truth look like, however?"[18] The answer to that is, "It looks like the statements you have just made about the metaphysical basis of absolute truth." Grondin continues to make assertions that are assumed to be absolutely true at the same time denying that there is absolute truth—or at least calling into question the legitimacy of the metaphysical foundation of absolute truth. In fact, Grondin goes on to declare, "There can never be an answer that all will acknowledge and accept."[19] This is, of course, certainly true. But it is also certainly irrelevant. Even if no one acknowledged and accepted an answer, this would not eliminate absolute truth, because it would be absolutely true that no one accepted it. Also, Grondin's own claim is a statement that is assumed to be absolutely true, and whether anyone wants to acknowledge and accept it or not is not the issue. The issue is whether it can be demonstrated that there is absolute truth, and Grondin has done an excellent job of doing just that.

Additionally, Soffer evaluates the "meaning" of the historicity thesis and concludes that "although it is clear that the thesis asserts [that] all knowledge is historically conditioned, the language of 'conditioning' is highly ambiguous, and some interpretations of 'conditioning' are wholly compatible with a traditional conception of objectivity." (Soffer 1992, p. 238). Soffer points out that contemporary concepts of mathematics could not have been formulated by more ancient cultures, but this does not obviate the objectivity of the current formulations. Soffer asserts,

> However, although historically conditioned, this knowledge remains "objective' in the following two senses: (1) once a given theorem becomes accessible at a local time and place, it can be reproduced with essentially the same meaning across a wide variety of present and future historical contexts (reproducibility of meaning); and (2) its validity is henceforth a matter of trans-historical, inter-subjective consensus: if mathematicians assign any truth-value at all to a given theorem, it is the same one (reproducibility of validity-assignment) (Soffer 1992, p. 238).

If by "historically conditioned" Soffer is employing Gadamer's notion of historical situatedness, then it is not reasonable to think that historically conditioned knowledge

remains objective. Gadamer's strong sense of historical conditioning is an effort to reject the positivistic approach to the human sciences, and yet this very rejection is supposed to follow from the historicity thesis. This is clearly circular, and begs the question. Gadamer spends the first lengthy part of this treatise in *Truth and Method* tracing the historical background that has led up to the monumental achievement of the realization of historicism. But, if there is no a-historical knowledge, why does Gadamer pretend that his report on these historical developments approach the truth about what really happened? Is Gadamer's own study of history not simply his own historically bound view from his own historical situatedness? Yet, he presents it as if this was *the* true and accurate account of the history. Gadamer's own rejection of positivistic approaches is asserted as if it were *the* historically *un*conditioned, objective truth about the human sciences. This is clearly self-referentially incoherent.

## 7. A Critique of the Current View

### 7.1. Misunderstanding Aristotle

In the Dictionary for Theological Interpretation of the Bible, Mark Bowald declares,

It is now generally acknowledged that the modern ideal of objectivity overreaches human capacities. Criticisms arise in the wake of the recognition of the person's location within certain untranscendable epistemological limitations, including time and space. Therefore, for the knower/reader there are only degrees of greater or lesser objectivity, which are always accompanied by a corresponding lesser or greater measure of subjectivity (Bowald 2005, p. 545).

It is impossible not to observe the self-defeating nature of such assertions. If, as Bowald claims, objectivity overreaches human capacities, one must wonder how Bowald knows this. Nothing-but statements always require more-than knowledge. If there are certain untranscendable epistemological limitations, then these limitations must certainly apply to Bowald, and, consequently, his observations about untranscendable epistemological limitations is simply his own statement within his own untranscendable epistemological limitations and are not objectively true. Of course authors who make such claims certainly hope that their readers take their own statements as objectively true and not merely the expressions of their own limitations.

Bowald believes he has support for his claims from the history of objectivity:

The history of the idea of "objectivity" has two phases. The first begins with ancient Greek philosophy and is dominant until the seventeenth century. In this phase questions about knowledge are overshadowed by the influence of Aristotle, whose *Physics* was a standard textbook for nearly two millennia. In this view the being or true essence of a thing determines how the thing is known: ontology dictates to epistemology the terms for objectivity.

In this arrangement, the "subject" refers to the true essence of the thing in itself. The "object" is the appearance of the thing as an expression of the essence that is presented to our mind or "intellect." The "problem" of knowledge is how persons should align their intellect with the "object" that is given to it. Aristotle assumes that the objective presentation of the thing is an adequate presentation of its true essence. The dilemma is not whether we have access to the true essence of a thing, but how we should respond to its givenness. "Objectivity," in this premodern view, is the proper aligning of the intellect to the object as the knower sorts out the essence of the thing given in the "objective" appearance. "Knowledge" is the successful alignment of the two (Bowald 2005, p. 544).

Besides the fact that Bowald talks about the history of objectivity as if he has an objective understanding beyond his own untranscendable epistemological limitations, he engages in a common misunderstanding of what Aristotle assumed. In the opening statements of *Peri Hermeneutica*, Aristotle uses the expression "passion of the soul."

16a 1 First we must establish what a name is and what a verb is; then what negation is and affirmation, and the enunciation and speech.

16a 3 Now those that are in vocal sound are signs of passions in the soul [paqhv-
mata thÆß yuchÆß], and those that are written are signs of those in vocal sound.

16a 5And just as letters are not the same for all men so neither are vocal sounds
the same;

16a 6 but the passions of the soul, of which vocal sounds are the first signs, are
the same for all [ta; aujta; paÆsi]; and the things of which passions of the soul are
likenesses are also the same.

16a 8 This has been discussed, however, in our study of the soul for it belongs to
another subject of inquiry.[20]

As is well known, *Peri Hermeneutica* is not a text on hermeneutics as this is usually
understood today, but these opening statements are important in understanding the ques-
tion of presuppositions and preunderstanding in interpretation and showing how Bowald
has misunderstood Aristotle. In the book, *Locke, Language and Early-Modern Philosophy*,
Hannah Dawson expresses the same kind of misunderstanding of Aristotle's introductory
statements as expressed by Bowald:

At the beginning of *De interpretatione* Aristotle had explained how words bridge
on to the world by means of the mind:

Words spoken are symbols or signs of affections or impressions of the soul;
written words are the signs of words spoken. As writing, so also is speech not the
same for all races of men. But the mental affections themselves, of which these
words are primarily signs, are the same for the whole of mankind, as are also the
objects of which those affections are representations or likenesses, images, copies.

This gobbet is the foundation of early-modern philosophy of language, containing
three claims that become, in the main, axiomatic. First, Aristotle lays down the
rarely contested law that words are conventional. Second, he declares that while
words, because they are purely conventional, differ between people, the concepts
(like the things) they signify are the same for all men. We will hear these two
claims being repeated, if sometimes tested, throughout this book. Now I look at
the third maxim that Aristotle dictates: words signify concepts which, in turn,
signify *and resemble* objects. While his followers debate heatedly about the matter,
they agree that concepts are integral to the signification of things. Moreover, they
tend to present these concepts not as obstacles, but as straightforward ways of
knowing those things (Dawson 2007, pp. 26–27).

Notice the translation of Aristotle's expression paqhvmata thÆß yuchÆß (*pathe4mata
te4s psyche4s*) as "affections or impressions of the soul." In his commentary on this text,
Thomas Aquinas specifically rejects this understanding of Aristotle's phrase. He explains
that Aristotle's phrase "passions of the soul," refers to the conceptions of the intellect:

5. When he speaks of passions in the soul we are apt to think of the affections of
the sensitive appetite, such as anger, joy, and the other passions that are custom-
arily and commonly called passions of the soul, as is the case in II *Nicomachean
Ethics*. It is true that some of the vocal sounds man makes signify passions of
this kind naturally, such as the groans of the sick and the sounds of other an-
imals, as is said in I *Politics*. Now the discussion concerns utterances that are
meaningful on account of human agreement. Thus, by the phrase *passions of the
soul* we must understand the concepts of the intellect, which nouns, verbs, and
expressions immediately signify, according to Aristotle's position. They cannot
immediately signify things, as is clear from their mode of signification; for the
noun *man* signifies human nature in abstraction from the singulars. Therefore, it
cannot immediately signify a singular human, whence the Platonists posited that
it should signify the separate Idea of Man. However, since this does not subsist in
the abstract in reality according to Aristotle's position, but is only in the intellect,

it was necessary for him to say that utterances immediately signify the concepts of the intellect and by their mediation the things.[21]

Aquinas points out that words immediately signify conceptions of the intellect (*intellectus conceptiones*), not things directly. This is because the word signifies the universal in the mind. If the word 'man' signified an individual man directly, then this would indicate that this individual man is *humanity*, and thus the word could not then be used to signify any other human being.

It is important to make the distinction between how Aquinas is using the term 'concept' when he talks about words immediately signifying conceptions of the intellect. For Aquinas, the concept is not the object of knowledge. The concept is not the "appearance of the thing," to use Bowald's expression. Rather, it is that *by which* the thing outside the mind, the *res extra animam*, is known. The concept is analogous to the judgment. The judgment is the act of combining or dividing. The judgment is not itself that about which the judgment is made. Rather, the judgement is the act of the mind in combining or dividing. Judgments are made about things. Similarly, the concept is not that which is known. Rather, the concept is that *by which* the *res extra animam* is known, as Aquinas explains:

> Now the one who understands may have a relation to four things in understanding: namely to the thing understood, to the intelligible species whereby his intelligence is made actual, to his act of understanding, and to his intellectual concept. This concept differs from the three others. It differs from the thing understood, for the latter is sometimes outside the intellect, whereas the intellectual concept is only in the intellect. Moreover the intellectual concept is ordered to the thing understood as its end, inasmuch as the intellect forms its concept thereof that it may know the thing understood. It differs from the intelligible species, because the latter which makes the intellect actual is considered as the principle of the intellect's act, since every agent acts forasmuch as it is actual: and it is made actual by a form, which is necessary as a principle of action. And it differs from the act of the intellect, because it is considered as the term of the action, and as something effected thereby. For the intellect by its action forms a definition of the thing, or even an affirmative or negative proposition. This intellectual concept in us is called properly a word, because it is this that is signified by the word of mouth. For the external utterance does not signify the intellect itself, nor the intelligible species, nor the act of the intellect, but the concept of the intellect by means of which [*qua mediante refertur ad rem*] it relates to the thing.[22]

The confusion concerning the term 'concept' is also illustrated in an article on semantics by Sten Ebbesen:

> Most late ancient philosophers were professedly Aristotelians in semantic matters. But the Stoa had left indelible marks on the philosophical tradition. The distinction between vocal significantia and intelligible significata was accepted and it was made Aristotelian by means of an interpretation of the beginning of *Peri Hermeneias* to the effect that Aristotle there says that words are signs of concepts, and concepts signs of things, this being equivalent to 'words are signs of things via concepts.' (Eschbach and Trabant 1983, p. 70).

In Porphyrian semantics, then, words—and in particular such as represent subject and predicate in a simple declarative sentence—signify concepts; and consequently the significatum of a whole sentence is a combination of concepts, a mental proposition, which in fact is nothing but a thought, as already indicated by Plato (*Sophist* 263–264). That is, the significata of sentences, the mental propositions, are psychological entities. (Eschbach and Trabant 1983, p. 70).

The statement of Aristotle to which Ebbesen refers does not claim that words are signs of concepts. Rather, Aristotle said words are signs of the "passions of the soul" [paqhvmata thÆß yuchÆß], and that the "passions of the soul" are the same for all [ta; aujta; paÆsi].[23] It can be easily verified that concepts, in the sense of ideas, or thoughts, or images, or notions,

are not the same for all. If what is known are concepts, how could Aristotle have declared that the passions of the soul are the same for all, especially since he goes on to assert that words are meaningful by convention and are not the same for all [Kai; w{sper oujde; gravmmata pa*Æ*si ta; aujtav, oujde; fwnai; aiJ aujtaiv]? The problem arises by Ebbesen's simple equation of the "passions of the soul" with concepts as psychological entities. Such an equation misses the metaphysical point of Aristotle's statement. The passions of the soul may be considered "psychological entities" in the sense that they exist in the mind of the knower, but they are not concepts. The passions of the soul are the forms of the entities that exist outside the soul that have come to exist in the soul. The forms of the extra mental entities come to exist in the mind of the knower and are these entities by virtue of their forms. They are the same for all because the entities that exist outside the soul, that is extra-mental reality, are the same entities regardless of the presuppositions and preunderstandings of anyone who uses words to signify them.

Although there were certainly conflicting interpretations of Aristotle's position on many points, Aristotle's own words show that he held the view that particulars in the world actually come to exist in the mind of the perceiver apart from any assumed larger framework of presuppositions or preunderstanding. It is simply not the case that for Aristotle the passions of the soul are "the appearance of the thing as an expression of the essence that is presented to our mind or 'intellect'" as Bowald claims. Rather the passion of the soul *is* the *res extra animam* that has come to exist in the mind by virtue of the form of the extra-mental object. As Aristotle explains,

> Generally, about all perception, we can say that a sense is what has the power of receiving into itself the sensible forms of things without the matter, in the way in which a piece of wax takes on the impress of a signet-ring without the iron or gold; what produces the impression is a signet of bronze or gold, but not *qua* bronze or gold: in a similar way the sense is affected by what is coloured or flavoured or sounding not insofar as each is what it is, but insofar as it is of such and such a sort and according to its form.[24]

The sense receives the sensible form from the entity outside the mind, and that form comes to exist in the mind. What is in the mind, as far as Aristotle was concerned, was not, as Bowald claims, "the appearance of the thing as an expression of the essence that is presented to our mind or 'intellect.'" Rather, what is in the mind is the *res extra animam*, the entity itself apart from its matter by virtue of its form. The passions of the soul are called a 'passions' (paqhvmata) because the soul passively receives the form of the *res extra animam*. And notice that Aristotle asserted that the passions of the soul "are the same for all" (ta; aujta; pa:si). Regardless of any larger framework of presuppositions or preunderstanding, according to Aristotle, extra-mental reality in the same for everyone.

No doubt some will object that the perceiver's linguistic community shapes the way he perceives the world. This is, of course, the linguistic relativity hypothesis. However, as John McWhorter, associate professor of English and comparative literature at Columbia University, has pointed out;

> Language does not shape thought in the way that one might reasonably suppose, nor do cultural patterns shape the way language is structured in the way that one might reasonably suppose. Rather, the way a language is structured is a fortuitously ingrown capacity. It is a conglomeration of densely interacting subsystems, wielded at great speed below the level of consciousness, endlessly morphing into new sounds and structures due to wear and tear and accreted misinterpretations, such that one day what was once Latin is now French and Portuguese.

> However, the perception capacity itself is the same regardless of the language. To be sure, a feeler, hooked into a certain patch of perception, enhances the speaker's sensitivity to the relevant phenomenon, and this book in no way denies the solid evidence for that. Yet the experiments in question have shown us that the enhancement qualifies as a passing flicker, that only painstaking experiment can

reveal, in no way creating a *different way of seeing the world* along the lines that a von Humboldt, von Treitschke, or anyone else would propose (McWhorter 2014, pp. 148–49).

Since the great Eskimo vocabulary hoax, the linguistic relativity hypothesis has been largely abandoned, at least as this was proposed by Franz Boas et al. Unfortunately, many evangelicals have not received the memo.

### 7.2. Misunderstanding the Role of Presuppositions

To the careful observer it may seem that some evangelical theorists may not have explicitly considered all of the implications of their current thinking on the role of presuppositions in interpretation. At times they seem to be arguing that *unacceptable* presuppositions can lead to *acceptable* conclusions, which issues in a realization that the unacceptable presuppositions must be revised. Since no interpretation is ever done apart from one's presuppositions, as the current thinking repeatedly warns, any given interpreter may come to the text with unacceptable presuppositions and employ these very unacceptable presuppositions in his initial encounter with the text. But current thinking says that somehow this interpreter arrives at the conclusion that at least some of his initial presuppositions are unacceptable and should be altered or eliminated. It would seem to follow that in this situation the interpreter would have arrived at the conclusion that the very presuppositions that were employed in the interpretation of the text are unacceptable, and that this conclusion was reached on the basis of these very same unacceptable presuppositions. In such a scenario, it would seem to be the case that unacceptable presuppositions have yielded an acceptable result. But if *unacceptable* presuppositions yield an *acceptable* understanding of the text, then what would be the evidence that would lead the interpreter to realize that his presuppositions were unacceptable? Is it reasonable to think that one's presuppositions would yield a result that would be contrary to the very presuppositions that yielded the result? Acceptable results would not imply that revisions were needed in the interpreter's presuppositions, for it was these very presuppositions that yielded the acceptable result.

Of course the standard response is that although some presuppositions may be unacceptable, others may not be, and it is on the basis of these other presuppositions that the interpreter discovers that he holds some presuppositions that are unacceptable. However, this approach does not resolve the problem. How does the interpreter discover from his interpretation which of his presuppositions are which? Do the conclusions from presuppositions *A* indicate that these are the acceptable ones, or do the conclusions from presuppositions *B* indicate that these are the acceptable ones? This distinction cannot be made on the basis of the conclusions of one's interpretation because the conclusions have been derived on the basis of the very presuppositions about which the distinction must be made.

However, most Evangelicals respond that interpretation is not a purely logical process (moving only from premises to conclusions), but that there are, at least for the Christian, other factors to be considered in the interpretive process, such as the work of the Holy Spirit or the impact of the living and dynamic Word of God. In the experience of the Christian interpreter, unacceptable or faulty presuppositions are often overturned or reworked as a result of the work of the Holy Spirit on the mind of the interpreter. But this is precisely the point. If it is the case that the Holy Spirit of God can override, in a sense, the unacceptable presuppositions of an interpreter [and this author believes that this can and often must happen], then the work of the Holy Spirit transcends the function of presuppositions and preunderstanding in the interpretive process as this is described in the literature. If the work of the Holy Spirit or the human interaction with the living and dynamic Word of God transcends the function of presuppositions in the interpretive process, then the account of the role of preunderstanding and presuppositions found in contemporary explanations needs to be modified to account for this factor, since neither the Holy Spirit nor the Word of God as living and dynamic can be readily subsumed under the explanations of preunderstanding and presuppositions that are popularly expounded today.

Why cannot the work of the Holy Spirit or the Word of God as living and dynamic be subsumed under the explanations of preunderstanding and presuppositions that are popularly expounded today? Because in the initial encounter with the Word of God one would need to have these presuppositions before one could get them. These presuppositions would come either from the World of God itself, or by special, direct intervention. Barring any direct, special intervention to a given interpreter today, the notion of the work of the Holy Spirit, for example, would either be something completely outside the preunderstanding of the interpreter, in which case this is contrary to the popular characterization, or the interpreter would need to have this presupposition already present before he initially comes to the text, which would be the source of this presupposition. So, the interpreter would need to have the presupposition before he could get it. But, of course, this is contradictory. If it is the work of the Holy Spirit that enables an interpreter to arrive at acceptable results, then it is not the case, as Bowald claims, that there are certain untranscendable epistemological limitations since the work of the Holy Spirit transcends our presuppositions and our epistemology.

### 8. Presuppositions and Truth-Claims

However, returning to the strictly philosophical analysis of the current explanation of the role of presuppositions in interpretation, a more serious problem in this methodology seems to present itself. The seriousness of this problem is seen in the fact that, if the criticisms are accurate, a failure to adjust the current account of the role of presuppositions could seriously weaken the hermeneutical systems of which this scenario seems to be a necessary presupposition.

As has been pointed out, presuppositions that are faulty have adverse implications for the truth-claims of which they are the presuppositions. The truth-claim of which a faulty presupposition is a necessary condition can no longer function in a system as part of the body of truths for that system, since that proposition either is false or has no truth-value; the truth-claims which presuppose faulty presuppositions become questionable themselves.

According to the popular view, the starting point of interpretation is within the preunderstanding of the interpreter. As Klien, et al. describe the process, "After an initial study of the Biblical text, that text performs a work on the interpreter. His or her preunderstanding is no longer what it was." ([Klein et al. 2017](), p. 114). Their extensive textbook provides the student of the Bible with sound principles of interpretation incorporating the most recent advances in genre analysis coupled with traditional grammatical-historical interpretation. However, their proposals about the presuppositions and preunderstanding of the interpreter do not provide the philosophical basis to sustain their hermeneutical system. The problem is not that their analysis is wrong. Rather, their characterization of the role of presuppositions and preunderstanding does not go far enough. They have not provided an adequate explanation to support the hermeneutic principles they espouse.

If it is the case, as KBH and most other theorists propose, that no one interprets anything without presuppositions, then upon approaching a given text, the interpreter will necessarily initially interpret the text through the grid of his presuppositions, or prejudices. This being the case, setting aside for the moment the supernatural factors mentioned above, the initial conclusions at which the interpreter arrives from his study of the text will be conclusions mediated through his always-already present presuppositions. According to the methodology proposed, the interpreter should then allow this understanding of the text to evaluate his presuppositions. The understanding that was initially gained in the interpreter's very first encounter with the text, however, was an understanding that was mediated through the very presuppositions which he now hopes will be changed by the text. However, it is through his always-already present presuppositions that he has gained this understanding in the first place.

Ultimately then, if it is one's interpretation, which has been made possible through his presuppositions, that the interpreter now employs to evaluate these very presuppositions, it seems that one's own presuppositions are employed to evaluate one's own presupposi-

tions. The problem with this scenario is that, apart from the intervention of someone or something from outside the spiral, it is not the text *qua* text that is performing a work on the interpreter's presuppositions, prejudices, and preunderstanding. Rather, it is the text *after* the interpreter has interpreted it through his presuppositions in his very first encounter with the text that is supposed to exercise this function. As Nietzsche put it, "Profoundly distrustful towards the dogmas of the theory of knowledge, I liked to look now out of this window, now out of that, though I took good care not to become finally fixed anywhere, indeed I should have thought it dangerous to have done so though finally: is it within the range of probabilities for an instrument to criticize its own fitness?"[25]

The text is interpreted through the presuppositional framework of the very interpreter whose preunderstanding is in need of modification. If it is the interpreted text that performs this work on the interpreter, then the limits of change of the interpreter's preunderstanding are preset by the interpreters always-already present presuppositional framework, and change at the starting point is not possible, apart from the intervention of something or someone that transcends the interpreter's presuppositional framework.

Let us attempt to illustrate this problem. If an interpreter comes to the biblical text with an antisupernatural bias as part of his preunderstanding, then his interpretation of the text that makes reference to supernatural events will be interpreted through this grid of antisupernaturalism. But, since his initial encounter with these texts is through this always-already present antisupernatural grid, his interpretation will, according to the popular methodology, yield conclusions that are supposed to instruct him that his antisupernatural presuppositions were unacceptable. But this does not seem to be a reasonable scenario. An interpreter who presupposes the impossibility of the supernatural will not naturally or logically interpret the text in such a way as to issue in a conclusion that would require the rejection of the very presuppositions through which he has interpreted this text. Unless someone or something outside the spiral intervenes, the prevailing view offers no basis upon which to expect such an interpreter to alter his perspective, and there is no basis for claiming that this should be possible. If, in fact, the interpretation of the text issues in the conclusion that his antisupernatural presuppositions were unacceptable, is it reasonable to think that it was these very antisupernatural presuppositions that yielded this interpretation? Would it not be more reasonable to think that some other assumptions of which the interpreter may not have been consciously aware, issued in the realization that an antisupernatural prejudice was unacceptable? But other such assumptions are not accounted for by the methodology proposed by contemporary theorists because this methodology talks strictly in terms of mutable frameworks of preunderstanding that cannot be universally true.

In fact, this is the strength of the relativist assertion. On what basis can it be argued or demonstrated that a thoroughgoing materialist would suddenly interpret the Bible from his materialist world view in such a manner that he would realize that he needed to change the very world view that has produced the interpretive conclusions? And yet, most Evangelicals would probably report that something very similar to this has happened either to them or someone they know personally. We were all enemies of God prior to our conversion. But, can we reasonably claim that our antagonistic world view somehow issued in an understanding of the Word of God that led us to jettison this antagonism and cast ourselves upon God's mercy? Such an event, by virtually universal testimony, was the act of God intervening in our lives, not a result of the spiral of understanding. Such a supernatural event must come from outside one's presuppositional framework and is not accounted for in the scenario of understanding.

Another problem arises also. If all aspects of our preunderstanding are mutable, subject to scrutiny and change, then there are no immutable, universal, or absolute presuppositions that provide the foundation upon which to verify a truth claim. How can we explain the fact that Born Again Christians seemed to have recognized the supernatural intervention as in fact supernatural and true? And, once God has altered an interpreter to his antagonistic world view, or at least a sufficient part of it, so as to remove the enmity between the

interpreter and God, does not this then become part of one's presuppositional framework? And now that this is part of our presuppositional framework, is it likewise subject to change and modification? If someone responds that this new aspect of our preunderstanding can grow and develop, that seems reasonable as far as it goes. But, would contemporary Evangelical theorists be willing to entertain the possibility that this new aspect of our preunderstanding can be jettisoned altogether? Of course someone will say something like this, "There is no reason to think that the text would lead the interpreter to make such a fundamental change." The problem with this response is that it comes dangerously close to objective truth and objective knowledge. If there is no objectivity, then what assurance can any theorist give that the text will not do precisely this in just the right circumstances? After all, contemporary theories do not make a distinction between universal presuppositions and mutable aspects of our preunderstanding, and they do not entertain the possibility of immutable and transcendent presuppositions.[26]

This problem of presuppositional circularity is the basis upon which contemporary theorists declare that objectivity in interpretation is not possible. Our analysis will focus on the proposals by Klein, Blomberg, and Hubbard because they provide the most extensive effort among Evangelicals to explain the hermeneutical spiral. KBH are cognizant of this implication of subjectivity and relativism from the current theory and flatly deny any charge that their view "simply jettisons all inductive assessment of the facts or data of the text and its situation." (Klein et al. 2017, p. 115). KBH seem to indicate that the appropriateness of the charge of subjectivity is obviated by the validity of their presuppositions. The self-referential problem in this assertion almost goes without comment. Who determines what presuppositions are valid? Would not one's presuppositions necessarily be employed in evaluating the validity of one's presuppositions. They seem to be saying that if their presuppositions are valid, then their interpretation is not subjective. The question is, Valid for whom? Who decides what is valid? They assert, "Recognizing the role of our preunderstanding does not doom us to a closed circle—that we find in a text what we want to find in a text—though that looms as an ever-present danger." (Klein et al. 2017, p. 115). How, then, does an interpreter insure that he does not succumb to the ever-present danger of circular subjectivity? According to KBH, the hermeneutical process is a spiral:

> The honest, active interpreter remains open to change, even to a significant transformation of preunderstanding. This is the hermeneutical *spiral*. Since we accept the Bible's authority, we remain open to correction by its message. There are ways to verify interpretations or, at least, to validate some interpretive options as more likely than others. It is not a matter of simply throwing the dice. There is a wide variety of methods available to help us find what the original texts most likely meant to their initial readers. Every time we alter our preunderstanding as the result of our interaction with the text we demonstrate that the process has objective constraints, otherwise, no change would occur; we would remain forever entombed in our prior commitments (Klein et al. 2017, p. 115).

A number of inconsistencies seem to be present in this explanation, however. First of all, KBH acknowledge a prior commitment to the Bible's authority. Yet they assert that unless they are willing to allow for the possibility of change in their preunderstanding, an aspect of which is their commitment to the Bible's authority, they would remain forever "entombed" in their prior commitments. Do they mean to imply, then, that they are willing to entertain the possibility that they will change their commitment to the Bible's authority? According to their own criteria, as honest, active interpreters, they must remain open to this significant transformation of their preunderstanding.

In other words, KBH seem on the one hand to be committed to the authority of the Bible as a doctrinal non-negotiable. This seems clear from such assertions as: "Secular historians may view the Bible only as a collection of ancient religious texts. To treat it as such—which often occurs in academia or among theologically liberal critics—cannot lead to valid conclusions about the religious value or significance of the Bible. The results are clearly 'sterile.' However, as authors we believe that the Bible is the divine word of

God. Only from that stance can we use our historical and critical methods and arrive at theologically meaningful and pertinent results." ([Klein et al. 2017](), p. 109).

However, on the other hand, their explanation of the role of presuppositions and preunderstanding indicates that there can be no presuppositional non-negotiables. In other words, for KBH, all presuppositions are subject to change. However, the very fact that KBH assert that the assumption that the Bible is simply a collection of ancient religious texts "cannot lead to valid conclusions about the religious value or significance of the Bible" indicates that an interpreter with these "liberal" presuppositions *cannot* arrive at a conclusion that will invalidate these presuppositions. If "liberal" presuppositions cannot lead to "valid" conclusions, then how can "non-liberal" presuppositions lead to "non-valid" conclusions? But, if their own non-liberal presupposition—biblical authority—cannot lead to non-valid conclusions, then how can their commitment to biblical authority be negotiable? And, if their commitment to biblical authority is non-negotiable, then what does it mean for them to declare that all presuppositions are changeable? It seems as if their own commitment to biblical authority is a presuppositional non-negotiable and therefore not subject to change. That being the case, they are simply wrong that all presuppositions are mutable.

For the moment, let us grant that their commitment to biblical authority is mutable, is capable of being changed or even abandoned. If they were to change their commitment to the Bible's authority on the basis of their interpretation of the biblical text, then the authority of the biblical text to correct their prior commitment to its authority is no longer authoritative. However, if the Bible is no longer authoritative, then why submit to its corrective instruction?

As interpreters who are committed to the Bible's authority, they claim to be "open to correction by its message." How do they come to understand this message? They come to understand the Bible's message by interpreting the various passages through which this message is communicated. But their interpretation is already filtered through their prior commitments. Consequently, their understanding of the message is an understanding of the message *as they interpret it through the framework of their preunderstanding*. If, then, they are understanding the message of the Bible through the framework of their preunderstanding, which includes their prior commitments, then ultimately it is their own prior commitments that are the means of correcting their prior commitments.

In an effort to escape what seems to be a vicious circle, KBH appeal to the concept of the Divine image in man to which they refer in a quote from Larkin: "because God made people in his own image they have the capacity to 'transcend preunderstanding, evaluate it, and change it.'" ([Klein et al. 2017](), p. 115). In this reference, Larkin asserts, "The mind of the interpreter, whether regenerate or unregenerate, evidences the fact that humans are made in God's image (Gen. 1:26)." ([Larkin 1993](), p. 299). Larkin's timely and important contribution to the field is specifically dedicated to opposing the rampant relativism in contemporary hermeneutic theory by reaffirming traditional hermeneutical principles. Larkin hits the relativist head-on with sound argumentation, tracing the hermeneutical and philosophical roots of the contemporary "crisis of biblical authority." ([Larkin 1993](), p. 18).

Nevertheless, Larkin's account of the role of presuppositions and preunderstanding needs further consideration in order to support his repudiation of relativism in hermeneutic theory. Note, for example, that viewing man as having been created in the image of God is a "prior commitment" characteristic of someone who already accepts the Bible's authoritative teaching on this matter. Indeed, the concept of the Divine image in man is a concept that derives from the biblical text as evidenced by Larkin's reference to Gen. 1:26. According to the prevailing view, however, in so far as it is derived from the biblical text, it is subject to the preconditional framework of the interpreter. Other interpreters may, and in fact do, hold contrary and even contradictory points of view about what constitutes the Divine image in man, or that such a thing exists. The fact that Larkin makes reference to the biblical passage from which his notion of the Divine image in man comes indicates that Larkin's

view derives from his interpretation of the biblical text which has been mediated through his already present preunderstanding.

According to the standards set by KBH, and assuming Larkin to be an honest, active interpreter, Larkin must at least be open to changing his interpretation of the nature of the Divine image because it is the product of his interpretation mediated through his mutable preunderstanding. However, if the notion of the Divine image in man is subject to change, then this would have serious implications for the hedge against circularity as KBH and Larkin are attempting to employ it.

KBH, Larkin, and others employ an appeal to the divine image as if this were a "brute fact" which obtains apart from their interpretations of any text. Indeed, they employ this interpretive conclusion as if it is an objective interpretation. If this is the case, and it cannot be ruled out a priori that this is in fact the case, then it must not be an aspect of one's preunderstanding as this is popularly defined, since preunderstanding is mutable and the concept of the Divine image in man is treated by these theorists as if it were immutable. Either it is a changeable aspect of one's preunderstanding, or it is not. If it is not a changeable aspect of one's preunderstanding, then it is either derived from the text as a product of one's interpretation, in which case it is the result of the application of one's own preunderstanding to the text and is therefore subject to change, or it is not. If it is not, then it is either an unchangeable foundation principle, or it is not. If it is not, then it is a changeable foundational principle; but this is just another name for one's preunderstanding.

But, if the image of God in man is an unchangeable foundation principle, then it is not the case that all presuppositions and every facet of one's preunderstanding is changeable. Along with the unchangeable logical principles of contradiction, excluded middle, and identity, there is this additional transcendental presupposition, the image of God in man.

So, it is either the case that all presuppositions are subject to change and significant transformation as part of one's preunderstanding, or some are not. Those presuppositions that are immutable cannot, then, be part of one's preunderstanding as it is popularly defined, since, according to the current wisdom, all preunderstanding is subject to change. The assertions of KBH, Larkin, Osborne, and others indicate that there may be presuppositions that in fact are not subject to change. According to these analyses, it may be the case that there are other necessary, unchangeable foundation principles upon which the interpretation of anything is based and which transcend all points of view and all perspectives. At least, this seems to be the way in which KBH, Larkin, Osborne, and others employ some of their own principles. For KBH it is the commitment to the Bible's authority. For Larkin, it is the Divine image in man. For Osborne it is placing ourselves "in front of the text" so that it can "address and if necessary change a presupposed perspective." (Osborne 2006, pp. 516–17). For virtually all theorists there is at least one unchangeable, foundational principle—the assumption that interpretation is always through the interpreter's framework of preunderstanding.

Similar observations can be made concerning the proposals of McCartney and Clayton. They assert that the interpreter's most basic presuppositions and his subordinate presuppositions should "be brought into line with that of the biblical writers." (McCartney and Clayton 1994, p. 17). But how does one come to know what are the presuppositions of the biblical writers? This would necessarily involve interpreting the biblical text. If it is necessary to interpret the biblical text in order to discover the presuppositions of the biblical writers, how can one know whether what one concludes from his interpretation are the actual presuppositions of the biblical writers, or whether they are merely the *perceived* view of the biblical author being mediated through the interpreter's own presuppositions? McCartney and Clayton acknowledge that "our presuppositions are going to influence how we look at our presuppositions," (McCartney and Clayton 1994, p. 17). but will they not also influence how one looks at the presuppositions of the biblical writers? They admit that the process of recognizing one's presuppositions is "an exceedingly difficult process," but they urge the process of attempting to recognize one's presuppositions in order to "evaluate whether and to what degree they are in harmony with those of the Bible;" further, they

insist that this is a lifetime project that "must continually be undertaken." (McCartney and Clayton 1994, p. 17). They go on to assert, "In fact, we should say that *the key to interpreting the Bible is to allow it to change and mold our presuppositions* into an interpretive framework compatible with the Bible." (McCartney and Clayton 1994, p. 17, Emphasis in original).

However, that they have not avoided the self-referential problem, and their description seems to involve them in the same kind of problem. Although McCartney and Clayton acknowledge the presence of this circularity when they admit that one's presuppositions are going to influence how one looks at his presuppositions, and although they claim that, in spite of this circularity, the task is not hopeless, they offer no substantive means of escape. They propose that the interpreter should examine his presuppositions to see whether they are "compatible with the Bible." How does the interpreter come to understand what is and what is not compatible with the Bible? Would not one have to interpret the Bible in order to discover what the Bible says? Ultimately, the interpreter's understanding of what is compatible with the Bible is the result of his interpretation of the Bible, which has already been influenced by his presuppositions since no interpretation is possible without them. To assert that our presuppositions must be brought into line with that of the biblical writers offers no hope since in order to know what the biblical writers said, we must interpret their texts, and in this view no one interprets any text apart from his presuppositional framework. Once again we are involved in what seems to be a vicious circle that never allows an objective grasp of the text, but allows the interpreter to know only the interpretation of the text through his always-already present framework of understanding.

Adjudication between those presuppositions that are considered acceptable and those that are considered unacceptable, following the methods that have been proposed by KBH, McCartney and Clayton, Larkin, and Osborne, is supposed to involve a critical examination of one's presuppositions in the effort to arrive at those that are acceptable. Yet apart from the intervention of someone or something outside the spiral, no escape from circularity is forthcoming. Strictly on the basis of the explanations of the role of presuppositions and preunderstanding, no methodology has been proposed that would enable the interpreter to transcend his perspective so as to evaluate his perspective. No unchangeable presuppositions have been offered that might serve as an immutable standard of judgment. No one seems to be able to propose a "view from nowhere" that would allow for a completely objective perspective. Examining one's presuppositions to discover those that are acceptable is ultimately, according to the current wisdom, simply a function of the always-already present framework. How can we legitimately argue that our own framework can examine itself in order to evaluate whether it is acceptable? To paraphrase Nietzsche's statement, "it is not within the range of probabilities for an instrument to criticize its own fitness."

Additionally, no Evangelical hermeneutician seems to have articulated a standard by which one can judge between what are acceptable and what are unacceptable presuppositions. This is not to say that Evangelicals do not believe that it is possible to adjudicate between good and bad interpretations. They certainly do offer extensive and effective explanations and examples of hermeneutical principles designed to lead an interpreter to a correct understanding of the text. However, on the level of presuppositions at the starting point of the hermeneutic process, short of the intervention of someone or something outside the spiral, none of the explanations offered allows for the possibility of adjudicating between presuppositions, nor do their methodologies support their own hermeneutical principles on this point. Such a standard cannot derive from the results of their interpretations of the biblical texts regardless of the assumed authority of the text. Even though one believes the Bible to be authoritative, it must still be interpreted, and if all interpretation is mediated through the mutable preunderstanding of the interpreter, which by their own assertions cannot be objective, no non-mediated result can ever be achieved, and thus no objective standard would seem to be accessible. As the relativist asserts, the results are results for me—not universally true or transcendently applicable.

This conclusion, however, is not considered by everyone to be a defect. In fact, some theorists have proposed that not only is no objective starting point necessary, no such starting point is either possible or even desirable. With reference to this circularity and the idea of a starting point, Peters asserts,

> The most important consequence of the circularity of understanding for hermeneutics is that there is no pure starting point for understanding because every act of understanding takes place within a finite historically conditioned horizon, within an already understood frame of reference. It is no longer a question of how we are to enter the hermeneutical circle, because human consciousness is always already in it. We understand only by constant reference to what we have already understood, namely, our past and anticipated experience. The experiencing and reflecting subject is never a *tabula rasa* upon which the understanding of raw experience inscribes its objective character; rather, all experience and reflection are the result of a confrontation between one's pre-understanding or even prejudice and new or perhaps strange objects. The inevitable presence of pre-understanding or prejudice is not necessarily the distortion of the meaning of an object by an arbitrary subject, rather, it is the very condition for any understanding at all (Peters 1974, p. 214).

On the one hand Peters asserts that there is in fact no "starting point" per se since "we are already always in it." On the other hand he disclaims any adverse effects on meaning. If, however, all interpretation is only mediated through one's preunderstanding, to say that there is no "distortion of the meaning" is nonsensical, since whatever meaning one obtains can never be evaluated with reference to an unmediated meaning against which the interpreted meaning can be judged as a distortion of *the* meaning. The only way any given interpretation can be disqualified as a "distortion of the meaning" is if someone possesses "the meaning" against which any other particular meaning is able to be judged. If all meanings are mediated meanings, then no meaning is "the" meaning. To assume that any particular meaning can be a distortion of "the" meaning not only assumes that "the" meaning exists, but that it is somehow accessible. This is what is meant by the "objective" meaning, which is precisely what Peters rejects.

Many Evangelicals have also concluded that objectivity is not possible. Peter Cotterell and Max Turner have contributed to the field by producing a volume that is supposed to incorporate advances in the field of linguistics. In their analysis of the role of the reader in interpretation, they assert, "Any confidence in the 'objectivity' of our findings must be further called into question by the frank recognition that many if not all scholars would be prepared to admit they are *ultimately* studying Paul (or Calvin or whomever) *in order to understand themselves and their God*." (Cotterell and Turner 1989, p. 70. Emphasis in original). They acknowledge that this notion implies the danger of misunderstanding and distortion, but they assert that this does not create a hopelessness in the interpreter's quest for the "discourse meaning." The influence of the interpreter and the lack of objectivity should, they claim, "generate the appropriate caution, both in respect of method and in the degree of certainty we attach to our 'conclusions.' We need fully to recognize that *our* reading of the letter to Philemon (or whatever), however certain we may feel it is what Paul meant, *is actually only a hypothesis*—our *hypothesis*—*about the discourse meaning*. It is the result of seeing certain aspects of the text and of providing what *we understand* to be the meaning that provides coherence to the evidence." (Cotterell and Turner 1989, p. 70. Emphasis in original).

The problem here is that if any interpretation of anything whatever is only a hypothesis, then no one possesses a non-hypothetical interpretation. If no non-hypothetical interpretation exists, then what is being used to measure everyone's interpretations by which the authors discover that all interpretations are only hypothetical? How can it be known that all interpretations are only hypothetical if a non-hypothetical interpretation from which to judge does not exist? Additionally, how can the reader, according to this scenario, obtain anything but a hypothetical interpretation of the analysis of Cotterell and

Turner. In fact, Cotterell and Turner present their analysis as if they expect the reader to have an objective interpretation of what they are asserting. They seem to have committed the fallacy of the lost distinction. They allow for no non-hypothetical interpretation, yet they globally assert that all interpretations are only hypothetical. Is this claim only hypothetical, or is it objectively true? Do these authors posses the non-hypothetical interpretation on the basis of which they can confidently assert that all interpretations are only hypothetical? If no one possess the measuring stick, then against what are they measuring? Cotterell and Turner do not consider these kind of problems.

Silva also discusses the question of the possibility of objectivity when he asserts, "In effect, this is what historical exegesis has had as its goal; total objectivity on the part of the interpreter so as to prevent injecting into the text any meaning other than the strictly historical one. But such objectivity does not exist. And if it did exist it would be of little use, because then we would simply be involved in a bare repetition of the text that takes no account of its abiding value." (Kaiser and Silva 1994, p. 244). Silva attributes this hermeneutical methodology to the influence of Immanuel Kant of whose work Silva says it was "undoubtedly a major turning point between modern thought and everything that preceded it." (Kaiser and Silva 1994, p. 241). Its effect was "so broad and so fundamental in character that no intellectual discipline could escape its impact—not even biblical interpretation, though it took a while for exegetes to figure out what was happening." (Kaiser and Silva 1994, p. 241).

According to Silva, this influence has resulted in a distinctive character attributable to contemporary hermeneutics, namely, "its emphasis on the *subjectivity* and *relativity* of interpretation." (Kaiser and Silva 1994, p. 241, Emphasis in original). In light of the influence of subjectivity and relativity in the hard sciences through such historical figures as Kant and Kuhn, Silva concludes,

> For one thing, these developments tell us that we probably have overestimated the differences between the sciences and the humanities. In both of these broad disciplines, the researcher is faced with a set of data that can be interpreted *only in the light of previous commitments*; in both cases, therefore, an interpreter comes—consciously or unconsciously—with a theory that seeks to account for as many facts as possible. Given the finite nature of every human interpreter, no explanation accounts for the data exhaustively (Kaiser and Silva 1994, p. 242, Emphasis in original).

If every interpreter comes with his own theory which ultimately cannot account for the data exhaustively, then it seems to follow that Silva's conclusion is a result of his own theory, which he has brought to the data of hermeneutics, and his own explanation does not account for the data exhaustively. In other words, since no explanation, including Silva's own explanation, accounts for the data exhaustively, it may be the case that there is some other starting point which might be objectively based which has not been accounted for by Silva or by the explanations of contemporary theorists. In fact, this proposal cannot be rejected on the basis of the notion of preunderstanding which has been advocated by contemporary theorists. It follows then, that some objective starting point may possibly exist for which the explanations of other theorists simply have not accounted, because, according to Silva, no explanation exhaustively accounts for all the data.

Either the starting point is an aspect of one's mutable preunderstanding or it is not. If it is, then the hermeneutical circle seems to be vicious, and there is no means to adjudicate between acceptable and unacceptable presuppositions. If it is not, then one's starting point is either a product of one's interpretation, or it is not. If it is, then we are back in the vicious hermeneutical circle. If it is not, then one's starting point is either an unchangeable foundation principle or it is not. If it is not, then the only options are that it is either a part of one's preunderstanding, or it is a result of one's interpretation. If, however, one's starting point is an unchangeable, foundation principle, then it must transcend the mutable aspects of one's preunderstanding, in which case it would seem to provide an objective starting point for interpretation.

## 9. Conclusions

Our theology derives from the interpretation of the text through our preunderstanding and presuppositions. Contemporary understanding of the role of preunderstanding and presuppositions is heavily influenced by modern philosophy, particularly the philosophies of Kant, Heidegger, and Gadamer. Consequently, the prevailing view of the role of presuppositions and preunderstanding in interpretation is ultimately self-defeating. While denying the possibility of objective interpretation theorists present their views as if they are objectively true, and they assume that their readers will objectively understand their claims. Proponents of the prevailing view claim that all interpretation takes place through the mutable presuppositions of the interpreter, but they assert this as an immutable presupposition of all interpretation. Purveyors of this view claim that no interpretation can be objective, yet they present their interpretation of interpretation as if it is objective and true.

The fact of the matter is, not only is objectivity possible, it is ultimately unavoidable. Not one theorist presents his views as if these views are only his own untranscendable epistemologically limited understanding that overreaches human capacities. Every one of these theorists discusses the topic as of he has an objective understanding of the nature of interpretation and understanding, and each one presents his interpretation of interpretation as if it is an undeniable, untranscendable brute fact. In other words, not one of these theorists can avoid propagating his view of interpretation and understanding as if it is objectively true.

No hermeneutic theorist asserts that presuppositions or preunderstanding is inevitable "for me." No one claims "I cannot come to a text without a framework of understanding." No one asserts, "My presuppositions are changeable, though perhaps yours are not." Rather, all theorists objectively claim that *everyone* comes to a text or to the world with an always-already present framework of understanding. All theorists objectively affirm that *no one* comes to a text or to the world with a *tabula rasa*. All theorists objectively declare that *everyone's* presuppositions are changeable (except of course the presupposition that everyone's presuppositions are changeable). Everyone objectively stipulates that *no one* can have absolute objectivity and that no one comes to the text as an objective observer. Scholars in the field of hermeneutics, Evangelicals and non-evangelicals alike, working from contrary and often contradictory perspectives and interpreting the same hermeneutic data, arrive at the same conclusion. The universal assertions about the preconditions of interpretation seem to be just as much an immutable, undeniable, unavoidable foundational presupposition as the foundationalism that everyone takes pains to reject. The fact that all theorists from all perspectives, from all world views, from all presuppositional frameworks conclude that no one can be objective is an instance of the very objectivity in interpretation that all these theorist adamantly deny.

**Funding:** This research received no external funding.

**Conflicts of Interest:** The author declares no conflict of interest.

## Notes

[1] (Bultmann 1957, pp. 409–17), published in English, "Is Exegesis without Presuppositions Possible?" *Encounter* 21 (Spring, 1960): pp. 194–200.

[2] (Heidegger 1967, §32.152). "Alle Auslegung bewegt sich ferner in der gekennzeichneten Vorstruktur. Alle Auslegung, die Verständnis beistellen soll, muß schon das Auszulegende verstanden haben. Man hat diese Tatsache immer schon bemerkt, wenn auch nur im Gebiet der abgeleiteten Weisen von Verstehen und Auslegung, in der philologischen Interpretation".

[3] (Immer 1873, p. 9). "Endlich (d) hat der Ausleger den Unterschied zwischen dem Werke des Schriftstellers und seiner eigenen Anschuung der Sache aufzuheben. Das Werk des Schriftstellers in seiner Gesammtheit ist eben auch eine historische Thatsache und will als solche behandelt werden. Dies wird um so besser gelingen, je mehr der Interpret von seinem eigenen Meinen oder Wissen abstrahiren und sich—vermöge seiner geschichtlichen Kenntniss—in den Schriftsteller und seine Zeit versetzen kan".

[4] (Bultmann 1957, p. 410), "Von der Frage der Voraussetzungslosigkeit im Sinne der Vorurteilslosigkeit ist die Frage der Voraussetzungslosigkeit in jenem anderen Sinn zu unterscheiden, und in diesem Sinne ist zu sagen: *voraussetzungslose Exegese kann es nicht*

*geben*. Daß es sie faktisch deshalb nicht gibt, weil jeder Exeget durch seine Individualität im Sinne seiner speziellen Neigungen und Gewohnheiten, seiner Gaben und seiner Schwächen bestimmt ist . . . " (translation by Schubert M. Ogden).

5    (Klein et al. 2017, p. 87), claim that "anyone who says that he or she has discarded all presuppositions and will only study the text objectively and inductively is either deceived or naive" See also, (Hasel 1985), p. 104, who argues, "The so-called 'empty head' principle whereby the investigator divests himself of all preconceived notions and opinions while approaching the subject to be studied in complete neutrality, is simply illusory". Moreover, Luiz Gustoavo da Silva Goncalves, "The Deconstructing of the American Mind: An Analysis of the Hermeneutical Implications of Postmodernism," in (Bauman and Hall 1995), p. 246, states, "Neutral exegesis does not exist," and Moisés Silva makes a similar claim when he asserts that "total objectivity on the part of the interpreter . . . does not exist". Silva, "Contemporary Approaches to Interpretation," p. 244.

6    (Gadamer 2010, p. 224)." . . . nicht so sehr unsere Urteile als unsere Vorurteile unser Sein ausmachen . . . Vorurteile sind nicht notwendig unberechtigt und irrig, so daß sie die Wahrheit verstellen. In Wahrheit liegt es in der Geschichtlichkeit unserer Existenz, daß die Vorurteile im wörtlichen Sinne des Wortes die vorgängige Gerichtetheit all unseres Erfahren-Könnens ausmachen. Sie sind Voreingenommenheiten unserer Weltoffenheit, die geradezu Bedingungen dafür sind, daß wir etwas erfahren, daß uns das, was uns begegnet, etwas sagt".

7    (Gadamer 2010, p. 274). "Erst solche Anerkennung der wesenhaften Vorurteilshaftigkeit alles Verstehens schärft das hermeneutische Problem zu seiner wirklichen Spitze zu".

8    (Schleiermacher 1974, pp. 141–42). "Der von Herrn Ast vorgetragene und nach manchen Seiten hin ziemlich ausgeführte hermeneutische Grundsaz, daß wie freilich das Ganze aus dem Einzelnen verstanden wird, so doch auch das Einzelne nur aus dem Ganzen verstanden werden könne, ist von solchem Umfang für diese Kunst und so unbestreitbar, daß schon die ersten Operationen nicht ohne AnWendung desselben zu Stande gebracht werden können, ja, daß eine große Menge hermeneutischer Regeln mehr oder weniger auf ihm beruhen".

9    (Heidegger 1967, §32, p. 152). "In jedem Verstehen von Welt ist Existenz mitverstanden und umgekehrt. Alle Auslegung bewegt sich ferner in der ekennzeichneten Vorstruktur. Alle Auslegung, die Verständnis beistellen soll, muß schon das Auszulegende verstanden haben".

10    (Heidegger 1967, §32, p. 153). "Aber *in diesem Zirkel ein vitiosum sehen und nach Wegen Ausschau halten, ihn zu vermeiden, ja ihn auch nur als unvermeidliche Unvollkommenheit »empfinden«, heißt das Verstehen von Grund aus mißverstehen*". Emphasis in original.

11    (Heidegger 1967, §32, p. 153). "Der »Zirkel« im Verstehen gehört zur Struktur des Sinnes, welches Phänomen in der existenzialen Verfassung des Daseins, im auslegenden Verstehen verwurzelt ist".

12    (Heidegger 1967, §32, p. 153). "*sondern in ihn nach der rechten Weise hineinzukommen*".

13    (Heidegger 1967, §32, p. 153). "*den Sachen selbst*".

14    (Gadamer 2010, pp. 271–72). "Daß jede Revision des Vorentwurfs in der Möglichkeit steht, einen neuen Entwurf von Sinn vorauszuwerfen, daß sich rivalisierende Entwürfe zur Ausarbeitung nebeneinander herbringen können, bis sich die Einheit des Sinnes eindeimiger festlegt; daß die Auslegung mit Vorbegriffen einsetzt, die durch angemessenere Begriffe ersetzt werden: eben dieses ständige Neu-Entwerfen, das die Sinnbewegung des Verstehens und Auslegens ausmacht, ist der Vorgang, den Heidegger beschreibt. Wer zu verstehen sucht, ist der Beirrung durch Vor-Meinungen ausgesetzt, die sich nicht an den Sachen selbst bewähren. Die Ausarbeitung der rechten, sachangemessenen Entwürfe, die als Entwürfe Vorwegnahmen sind, die sich >an den Sachen< erst bestätigen sollen, ist die ständige Aufgabe des Verstehens".

15    (Gadamer 2010, p. 350). "Mann kann die innere Widersprüchlichkeit eines jeden Relativismus noch so klar aufweisen–es ist schon so, wie Heidegger es ausgesprochen hat: alle diese siegreichen Argumentationen haben etwas vom Überrumpelungsversuch an sich. So überzeugend sie scheinen, so sehr verfehlen sie doch die eigentliche Sache. Man behält recht, wenn man sich ihrer bedient, und doch sprechen sie keine überlegene Einsicht aus, die fruchtbar wäre. Daß die These der Skepsis oder des Relativismus selber wahr sein will und sich insofern selber aufhebt, ist ein unwiderlegliches Argument. Aber wird damit irgend etwas geleistet? Das Reflexionsargument, das sich derart als siegreich erweist, schlägt vielmehr auf den Argumentierenden zurück, indem es den Wahrheitswert der Reflexion suspekt macht. Nicht die Realität der Skepsis oder des alle Wahrheit auflösenden Relativismus wird dadurch getroffen, sondern der Wahrheitsanspruch des formalen Argumentierens überhaupt".

16    (Soffer 1992, p. 237). Soffer employs the term 'historicity thesis' to refer to the same concept referred to herein by the term 'historicism'.

17    (Grondin 2001, p. 24). "Die philosophische Leistung der Hermeneutik liegt vielleicht weniger in einer Lösung seines Problems als in einem Abschied vom Historismus. Bei Heidegger und Gadamer wird der Historismus sozusagen auf sich selbst angewendet und damit in seiner eigenen Geschichtlichkeit, nämlich in seiner geheimen Metaphysikabhängigkeit sichtbar gemacht. Denn die dogmatische These, alles sei relativ, kann nur vor dem Horizont einer unrelativen, absoluten, überzeitlichen, metaphysischen Wahrheit Sinn geben. Nur am Maßstab einer für möglich gehaltenen absoluten Wahrheit kann eine Meinung als bloß relativ gelten".

18    (Grondin 2001, p. 24). "Wie sieht aber diese absolute Wahrheit positiv aus?"

19    (Grondin 2001, p. 24)."Eine allgemeinbefriedigende, also von allen anerkannte Antwort wurde nie gegeben".

20    (Aristotelis 1927, 16a1–8). "16a1. PrwÆton deiÆ qevsqai, tiv o[noma kai; tiv rJhÆma, e[peita, tiv ejstin ajpovfasiß kai; katavfasiß, kai; ajpovfansiß kai; lovgoß. 16a3 [Esti me;n ou\n ta; ejn thÆ/ fwnhÆ/ twÆn ejn th/Æ yuchÆ/ paqhmavtwn suvmbola, kai; ta;

grafovmena twÆn ejn thÆ/ fwnhÆ/. 16a5 Kai; w{sper oujde; gravmmata paÆsi ta; aujtav, oujde; fwnai; aiJ aujtaiv: 16a6 w|n mevntoi tauÆta shmeiÆa prwvtwß, ta; aujta; paÆsi paqhvmata thÆß yuchÆß, kai; w|n tauÆta oJmoiwvmata, pravgmata h[dh taujtav. 16a8 peri; me;n ou\n touvtwn ei[rhtai ejn toiÆß peri; yuchÆß: a[llhß ga;r tauÆta pragmateivaß". "Primum oportet ponere, quid sit Nomen et quid Verbum, deinde, quid sit Negatio et Affirmatio, et Enuntiatio et Oratio. Sunt vero voce emissa passionum animæ signa, et scripta, eorum quæ voce emittuntur. Et quemadmodum nec literæ omnibus eædem sunt, *ita* nec voces omnibus eædem: quorum tamen hæc signa primo sunt, ea omnibus sunt eædem passiones animæ; et quorum hæc imagines sunt, ea quoque *omnibus* sunt res eædem. De his quidem dictum est in libris de Anima: sunt enim alienæ tractationis".

[21] (Aquinatis 1955, 5.12). "Circa id autem quod dicit, *earum quae sunt in anima passionum*, considerandum est quod *passiones* animae communiter dici solent appetitus sensibilis affectiones, sicut *ira*, *gaudium* et alia huiusmodi, ut dicitur in II *Ethicorum*. Et verum est quod huiusmodi passiones significant naturaliter quaedam voces hominum, ut *gemitus* infirmorum, et aliorum animalium, ut dicitur in I *Politicae*. Sed nunc sermo est de vocibus significativis ex institutione humana; et ideo oportet passiones animae hic intelligere intellectus conceptiones, quas nomina et verba et orationes significant immediate, secundum sententiam Aristotelis. Non enim potest esse quod significent immediate ipsas res, ut ex ipso modo significandi apparet: significat enim hoc nomen homo naturam humanam in abstractione a singularibus. Unde non potest esse quod significet immediate hominem singularem; unde Platonici posuerunt quod significaret ipsam ideam hominis separatam. Sed quia hoc secundum suam abstractionem non subsistit realiter secundum sententiam Aristotelis, sed est in solo intellectu; ideo necesse fuit Aristoteli dicere quod voces significant intellectus conceptiones immediate et eis mediantibus res".

[22] (Aquinatis 1883, VIII.1). "Intelligens autem in intelligendo ad quatuor potest habere ordinem: scilicet ad rem quæ intelligitur, ad speciem intelligibilem, qua fit intellectus in actu, ad suum intelligere, et ad conceptionem intellectus. Quæ quidem condeptio a tribus prædictis differt. A re quidem intellecta, quia res intellecta est interdum extra intellectum; conceptio autem intellectus non est nisi in intellectu; et iterum conceptio intellectus ordinatur ad rem intellectam sicut ad finem; propter hoc enim intellectus conceptionem rei in se format ut rem intellectam cognoscat. Differt autem a specie intelligibili: nam species intelligibilis, qua fit intellectus in actu, consideratur ut principium actionis intellectus; cum omne agens agat secundum quod est in actu: actu autem fit per aliquam formam, quam oportet esse actionis principium. Differt autem ab actione intellectus: quia prædicta conceptio consideratur ut terminus actionis, et quasi quoddam per ipsam constitutum Intellectus enim sua actione format rei definitionem, vel etiam propositionem affirmativam seu negativam. Hæc autem conceptio intellectus in nobis proprie verbum dicitur: hoc enim est quod verbo exteriori significatur: vox enim exterior neque significat ipsum intellectum, neque speciem intelligibilem, neque actum intellectus; sed intellectus conceptionem qua mediante refertur ad rem".

[23] (Aristotelis 1927, 16a 1–8). "16a1. PrwÆton deiÆ qevsqai, tiv o[noma kai; tiv rJhÆma, e[peita, tiv ejstin ajpovfasiß kai; katavfasiß, kai; ajpovfansiß kai; lovgoß. 16a3 [Esti me;n ou\n ta; ejn thÆ/ fwnhÆ/ twÆn ejn th/Æ yuchÆ/ paqhmavtwn suvmbola, kai; ta; grafovmena twÆn ejn thÆ/ fwnhÆ/. 16a5 Kai; w{sper oujde; gravmmata paÆsi ta; aujtav, oujde; fwnai; aiJ aujtaiv: 16a6 w|n mevntoi tauÆta shmeiÆa prwvtwß, ta; aujta; paÆsi paqhvmata thÆß yuchÆß, kai; w|n tauÆta oJmoiwvmata, pravgmata h[dh taujtav. 16a8 peri; me;n ou\n touvtwn ei[rhtai ejn toiÆß peri; yuchÆß: a[llhß ga;r tauÆta pragmateivaß". "Primum oportet ponere, quid sit Nomen et quid Verbum, deinde, quid sit Negatio et Affirmatio, et Enuntiatio et Oratio. Sunt vero voce emissa passionum animæ signa, et scripta, eorum quæ voce emittuntur. Et quemadmodum nec literæ omnibus eædem sunt, *ita* nec voces omnibus eædem: quorum tamen hæc signa primo sunt, ea omnibus sunt eædem passiones animæ; et quorum hæc imagines sunt, ea quoque *omnibus* sunt res eædem. De his quidem dictum est in libris de Anima: sunt enim alienæ tractationis".

[24] (Aristotelis 1927, II.12.424a.460). Kaqovlou de; peri; pavshV aijsqhvsewV dei: laqei:n o{ti hJ me;n ai[sqhsivV ejsti to; dektiko;n tw:n aijsqhtw:n eijdw:n a[neu th:V u{lhV, o|on oJ khro;V tou: daktulivou a[neu tou: sidhvrou kai; tou: crusou: devcetai to; shmei:on. Lambavnei de; to; crusou:n h] to; calkou:n shmei:on, ajll j oujc h|| / cruso;V, h] calcovV. JomoivwV de; kai; hJ ai[sqhsiV eJkavstou uJpo; tou: e[contoV crw:ma, h] cumo;n, h] yovfon pavscei, ajll j oujc h|| / e{caston ejkeivnwn levgetai, ajll j h/|| toiondi;, kai; kata; to;n lovgon. "Hoc autem universaliter accipere de omni sensu oportrl, sensum quidem id esse quod sensibiles formas sine materia suscipere potest, perinde atque annuli signum sine feno vel auro suscipit cera. Suscipit autem aeneum vel aureum signum, sed non ut aurum aut aes. Simili patitur et unhiMcu jusque sensus modo ab eo quod habet colorem, aut sapmem, aut sonum, sed non ut unumquodque illorum dicitur, sed ut est tale quid, et secundum rationem". Translated by J. A. Smith.

[25] (Nietzsche 1926, 410, p. 286). "Gegen die erkenntnisstheoretischen Dogmen tief misstrauisch, liebte ich es, bald aus diesem, bald aus jenem Fenster zu blicken, hüte mich, mich darin festzusetzen, hielt sie für schädlich, —und zuletzt: ist es wahrscheinlich, dass ein Werkzeug seine eigene Tauglichkeit kritisiren kann??"

[26] It is highly probable that evangelical theorists would gladly affirm the intervention of God to change the interpreter's mind, but they do not explicitly include this as a formal aspect of their methodology as it touches on the role of preunderstanding and presuppositions in interpretation.

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
