# Peer review of "Preunderstanding, Presuppositions and Biblical Interpretation"

_religions, doi:10.3390/rel13121206_

Round 1

Reviewer 1 Report

1. Overall this is a well-written paper. It presents a comprehensive and profound discussion and engagement with the philosophies/works of well-known philosophers of hermeneutics and a host of secondary sources pertinent to the topic. This is a very commendable research work. It shows the grasp of the author of these sometimes complex concepts of hermeneutics, specifically biblical hermeneutics, more so when dealing with thinkers like Kant, Heidegger, and Gadamer, Aristotle, Thomas Aquinas, Nietzsche. The author also engaged other thinkers/theorists in the Evangelical traditions and in hermeneutics to articulate his position and provide a more dialogical approach in his presentation. The rich citations and references clearly show the extent of the knowledge that the author has on the topic he is working on.

2. While this paper could be an excellent source of related literature on biblical hermeneutics, it does not have a very clear abstract. Without a clear abstract, the direction or focus of the paper is somehow obscured. In the Introduction, the author states that “no one interprets the text apart from his preunderstanding and presuppositions. The question of the role of preunderstanding and presuppositions in understanding has become a dominate (sic; dominant) topic in contemporary biblical hermeneutics.” Then it mentions philosophers like Kant, Heidegger, and Gadamer and other thinkers, theorists, and authors of a popular hermeneutics textbook on biblical interpretation. Then in the succeeding sections, it discusses the contents of the paper. The main contention/thesis may have been articulated and discussed in the main text, (in the Conclusion that is well spelled out; if I get it right, it is that our interpretation of the text is grounded on our preunderstanding and presuppositions), but this must be spelled out clearly in the abstract so that the readers can know the thesis or the position that the paper wants to convey or present.

3. I find the paper very substantial in its theoretical analysis of biblical interpretation based on the hermeneutical tradition. However, it would have grounded the paper on the process or application of preunderstanding and presuppositions if it also provided how these two are applied in biblical texts, accounts, or stories. While there is a mention of Isaiah and the Holy Spirit, such is inadequate. How do preunderstanding and presupposition figure in our interpretation, for example, of the role of the Virgin Mary in redemption or the Book of Revelation?

4. Lastly, I find this life in the Conclusion quite intriguing: “all theorists objectively claim that everyone comes to a text or to the world with an always-already present framework of understanding.” Two questions: What would be the basis of objectivity if every theorist already has a framework of understanding? Does the author argue against a realist perspective – that there is an objective reality or content of the Bible? The author may consider or not these questions.

Author Response

  1. Thank you for your kind comments.
  2. You are certainly correct about the abstract. As you have pointed out, I am not very good at writing abstracts.
  3. You are also correct about the lack of references to the biblical text. The only reason for this deficiency is the effort not to add several more pages to the article. Perhaps such material could have been added in such a way so as not to extend the length. I very much appreciate this observation.
  4. Concerning the questions raised, I have written on these very questions. Again I did not want to extend the length of the article. I have been accused of being verbose, and this is certainly the case.

Reviewer 2 Report

Even though this paper does not present or argue for a specific hermeneutical approach to the Bible, the argument for the reality of the presence of presuppositions in all methods of interpretation is significant and well argued. Apart from a few spelling errors, the article seems to be well researched and argued.

Author Response

Thank you for your generous comments.

Round 2

Reviewer 1 Report

I think the Abstract is good. 

The length of the article is justifiable, given the scope of the topic and the extent of the research.